# How Does Preconditioning Guide Feature Learning in Deep Neural Networks?

## Abstract

Preconditioning is widely used in machine learning to accelerate convergence on the empirical risk, yet its role on the expected risk remains underexplored. In this work, we investigate how preconditioning affects feature learning and generalization performance. We first show that the input information available to the model is conveyed solely through the Gram matrix defined by the preconditioner's metric, thereby inducing a controllable spectral bias on feature learning. Concretely, instantiating the preconditioner as the $p$-th power of the input covariance matrix and within a single-index teacher model, we prove that in generalization, the exponent $p$ and the alignment between the teacher and the input spectrum are crucial factors. We further investigate how the interplay between these factors influences feature learning from three complementary perspectives: (i) Robustness to noise, (ii) Out-of-distribution generalization, and (iii) Forward knowledge transfer. Our results indicate that the learned feature representations closely mirror the spectral bias introduced by the preconditioner—favoring components that are emphasized and exhibiting reduced sensitivity to those that are suppressed. Crucially, we demonstrate that generalization is significantly enhanced when this spectral bias is aligned with that of the teacher.

## 1 Introduction

In machine learning, preconditioning has traditionally aimed to accelerate convergence and reduce computational costs during training. By utilizing second-order information, such as the Hessian (Le-Cun et al., 2002) or the Fisher Information Matrix (FIM) (Amari, 1998), researchers have sought to navigate ill-conditioned loss landscapes (Li et al., 2018) more efficiently. Yet, the ultimate objective extends beyond convergence speed to achieving strong generalization on unseen data. This generalization fundamentally hinges on the nature of the feature representations learned during training (Bengio et al., 2013; Yosinski et al., 2014; LeCun et al., 2015; Damian et al., 2022). Although much of the literature has focused on accelerating convergence on the empirical risk, the impact on the expected-risk performance has often been underexplored (Dauphin et al., 2014; Keskar et al., 2016; Wilson et al., 2017). This motivates the question: how does preconditioning affect solution quality—namely, *what kinds of features are captured and how they contribute to generalization*?

The answer to this question remains controversial and has yet to be systematically established. One perspective suggests that preconditioning, particularly with second-order methods, can be detrimental. This view is supported by arguments that such methods converge to sharp solutions that generalize poorly (Keskar et al., 2016; Shin et al., 2025), or that they are equivalent to data whitening, which can discard useful information (Wadia et al., 2021). An opposing view, however, highlights the benefits, showing that preconditioning can enhance robustness to label noise (Amari et al., 2021), stabilize feature learning (Ishikawa & Karakida, 2024), and better handle anisotropic data (Zhang et al., 2025). It is also argued, from a middle-ground perspective and aligning with the No Free Lunch theorem in optimization (Wolpert & Macready, 1997), that the effectiveness of a preconditioner is not universal, but rather depends on the alignment of its inherent inductive bias with the specific structure of the task (Amari et al., 2021).

Despite these insights, a systematic understanding of the interplay between preconditioning and feature learning in deep learning is still lacking. In this paper, we aim to systematize the relationship between preconditioning and generalization in deep learning. First, we extend the arguments of

Wadia et al. (2021) to a general preconditioning framework. We demonstrate that neuron-wise preconditioning in the first layer determines the similarity metric of the Gram matrix, which uniquely conveys input data information to the model during both training and inference. In other words, the model can only access input information through the Gram matrix defined by the preconditioner's metric; the preconditioner directly shapes this similarity space, thereby directly influencing learning and inference. Next, we instantiate this insight by approximating the first-layer preconditioner as the $p$-th power of the input covariance matrix. We show that the exponent $p$ determines the similarity geometry of the Gram matrix, selectively emphasizing features with specific variance levels. Specifically, in this geometry, a larger $p$ makes high-variance features more dominant, while a smaller $p$ emphasizes low-variance components. Consequently, by considering a single-index teacher model, we conclude that, for a fixed input spectrum, test-time generalization is only governed by both the preconditioner's power $p$ and the degree of alignment between the teacher and the input spectrum, as well as the level of label noise.

We further elucidate how the interplay between the preconditioner's power $p$ and the alignment between the teacher and the input spectrum governs model generalization, examining this relationship from three perspectives: (i) **Robustness to noise.** We construct synthetic datasets where a teacher model is aligned with either high- or low-variance components. By training a two-layer MLP with preconditioners based on both the exact covariance matrix and an approximate Hessian, we find that the model becomes sensitive to the variance components emphasized by $p$, and that the value of $p$ yielding the most robust generalization is the one that matches the variance components to which the teacher is aligned. (ii) **Out-of-distribution (OOD) generalization.** The trend observed in (i) becomes particularly salient under a correlation shift, where the data contains both invariant and spurious features. We find that OOD generalization is improved when the power $p$ is chosen to emphasize the variance components associated with invariant features while suppressing those associated with spurious features. This is confirmed by comparing various optimizers on an MNIST-based task where small-variance Gaussian noise is added. (iii) **Forward knowledge transfer.** We connect our framework to transfer and continual learning, where leveraging knowledge from past tasks is crucial. We argue that using an optimizer with a strong spectral bias on a source task can inadvertently discard information vital for subsequent tasks, thereby degrading transfer/continual learning performance. Therefore, we show that setting $p = -1$, which transparently incorporates all spectral components during learning on the source task, is preferable for improving transferability to future tasks. We confirm this phenomenon under the same experimental settings as those used in (i). The code is available at https://anonymous.4open.science/r/preconditioning-feature-learning-9FC3.

**In summary, our main contributions are as follows:**

- We provide a theoretical result: when training deep models with preconditioning, all input information that can affect feature learning is extracted solely through the Gram matrix defined by the similarity geometry of the preconditioning matrix (Sec. 3.1).

- We instantiate this result to preconditioning based on the $p$-th power of the input covariance matrix and show that, in the induced Gram matrix, larger $p$ gives greater influence to high eigenvalue components whereas smaller $p$ gives greater influence to low eigenvalue components. Under a single-index teacher model, when the data spectrum is fixed, the model's generalization is determined only by $p$ and the alignment between the teacher and the input spectrum (and label noise) (Secs. 3.2 and 3.3).

- We further examine how the relationship between the preconditioner power $p$ and the alignment between the teacher and the input spectrum impacts generalization through three lenses: robustness to noise, OOD generalization, and forward knowledge transfer, and we find that in feature learning the components emphasized by the preconditioner are preferentially learned whereas the suppressed components are downweighted; generalization improves when this spectral bias is aligned with the teacher's (Sec. 4).

## 2 RELATED WORK

**Second-order optimization.** In large-scale deep learning, training efficiency is critical, and second-order optimization has long been pursued to speed up convergence. These methods reshape the gradient via preconditioning derived from the loss Hessian, but computing the Hessian and its

inverse is prohibitively expensive for modern networks; consequently, practical approaches rely on approximations. Representative examples include limited-memory quasi-Newton methods such as L-BFGS (Liu & Nocedal, 1989), Hessian-free optimization solved with matrix–vector products and conjugate gradients (Martens et al., 2010), Shampoo (Gupta et al., 2018), and lightweight curvature estimators that exploit diagonal or low-rank structure such as AdaHessian (Yao et al., 2021) and Sophia (Liu et al., 2024). Natural gradient descent (Amari, 1998) replaces the Hessian with the inverse FIM, with common approximations including diagonal, block-diagonal, and Kronecker factorizations (Kingma & Ba, 2014; Martens & Grosse, 2015) to ease the computational burden.

**Preconditioning and generalization.** It is well known that the choice of optimizer can materially affect generalization (Luo et al., 2019; Pascanu et al., 2025). A first line of evidence suggests that second-order preconditioning may be detrimental: motivated by the observation that flatter solutions tend to generalize better (Keskar et al., 2016), sharpness-aware methods explicitly bias training toward flat minima (Foret et al., 2021; Shin et al., 2025). In linear models, preconditioning with second-order information reduces to data whitening, which has been argued to be harmful because it can reduce the information available for generalization (Wadia et al., 2021); nevertheless, in NLP, whitening word or sentence embeddings has sometimes improved performance (Su et al., 2021; Yokoi et al., 2024). On the other hand, a growing body of work highlights benefits of preconditioning: natural gradient exhibits optimal robustness to label noise (Amari et al., 2021), certain parameterizations yield more stable feature learning than gradient descent (GD) (Ishikawa & Karakida, 2024), and K-FAC can outperform GD when the input distribution is anisotropic (Zhang et al., 2025). Consistent with No Free Lunch theorem in optimization (Wolpert & Macready, 1997), it is indicated that alignment between the preconditioner and the target labels is essential for good generalization in overparameterized linear models (Amari et al., 2021); under covariate shift, the optimal preconditioner can be characterized and varies with the target distribution (Liu et al., 2025).

## 3 THEORETICAL ANALYSIS

**Setup.** We consider a data distribution $\mu$ on the product space $\mathcal{X} \times \mathcal{Y}$, with $\mathcal{X} \subseteq \mathbb{R}^{d_x}$ and $\mathcal{Y} \subseteq \mathbb{R}^{d_y}$. The training sample is $\mathcal{D}_{\text{train}} = \{(\boldsymbol{x}_i, \boldsymbol{y}_i)\}_{i=1}^N$ with $(\boldsymbol{x}_i, \boldsymbol{y}_i) \overset{\text{i.i.d.}}{\sim} \mu$, and stacking inputs and labels columnwise gives $\boldsymbol{X}_{\text{train}} = [\boldsymbol{x}_1, \ldots, \boldsymbol{x}_N] \in \mathbb{R}^{d_x \times N}$ and $\boldsymbol{Y}_{\text{train}} = [\boldsymbol{y}_1, \ldots, \boldsymbol{y}_N] \in \mathbb{R}^{d_y \times N}$, and we assume $d_x \leq N$. We study a model $f(\cdot; \boldsymbol{W}_1, \boldsymbol{\theta}_{2:L}) : \mathbb{R}^{d_x} \rightarrow \mathbb{R}^{d_y}$ whose first layer is fully connected with column-vector convention $\boldsymbol{z} = \boldsymbol{W}_1^\top \boldsymbol{x} \in \mathbb{R}^{d_h}$ for $\boldsymbol{W}_1 \in \mathbb{R}^{d_x \times d_h}$, and the remaining layers $g_{2:L}(\cdot; \boldsymbol{\theta}_{2:L}) : \mathbb{R}^{d_h} \rightarrow \mathbb{R}^{d_y}$ satisfy $f(\boldsymbol{x}; \boldsymbol{W}_1, \boldsymbol{\theta}_{2:L}) = g_{2:L}(\boldsymbol{W}_1^\top \boldsymbol{x}; \boldsymbol{\theta}_{2:L})$ for $\boldsymbol{\theta}_{2:L} \in \mathbb{R}^{d_\theta}$. Let $\ell : \mathbb{R}^{d_y} \times \mathbb{R}^{d_y} \rightarrow \mathbb{R}$ be a per-sample loss; the empirical risk minimized during training is $L(\boldsymbol{X}, (\boldsymbol{W}_1, \boldsymbol{\theta}_{2:L})) = \frac{1}{N} \sum_{i=1}^N \ell(f(\boldsymbol{x}_i; \boldsymbol{W}_1, \boldsymbol{\theta}_{2:L}), \boldsymbol{y}_i)$.

### 3.1 THE IMPACT OF PRECONDITIONED SIMILARITY GEOMETRY ON FEATURE LEARNING

We first consider the following neuron-wise preconditioned update on the first layer. At step $t$, the update

$$\boldsymbol{W}_1^{(t+1)} = \boldsymbol{W}_1^{(t)} - \eta \, \boldsymbol{P}^{(t)} \frac{\partial L^{(t)}}{\partial \boldsymbol{W}_1^{(t)}} \tag{1}$$

$$= \boldsymbol{W}_1^{(t)} - \eta \, \boldsymbol{P}^{(t)} \, \boldsymbol{X}_{\text{train}} \left( \frac{\partial L^{(t)}}{\partial \boldsymbol{Z}_{\text{train}}^{(t)}} \right)^\top, \tag{2}$$

where $\boldsymbol{P}^{(t)} \in \mathbb{R}^{d_x \times d_x}$ is positive semi-definite, $\boldsymbol{Z}_{\text{train}}^{(t)} = \boldsymbol{W}_1^{(t)\top} \boldsymbol{X}_{\text{train}}$, and $L^{(t)} = L(\boldsymbol{X}_{\text{train}}, (\boldsymbol{W}_1^{(t)}, \boldsymbol{\theta}_{2:L}^{(t)}))$. The remaining parameters are updated by $\boldsymbol{\theta}_{2:L}^{(t+1)} = \boldsymbol{\theta}_{2:L}^{(t)} - \eta \, \boldsymbol{Q}^{(t)} \frac{\partial L^{(t)}}{\partial \boldsymbol{\theta}_{2:L}^{(t)}}$ with a positive semi-definite preconditioner $\boldsymbol{Q}^{(t)} \in \mathbb{R}^{d_\theta \times d_\theta}$.

Define the preconditioned Gram at step $t$ as $\boldsymbol{G}_P^{(t)} := \boldsymbol{X}_{\text{train}}^\top \boldsymbol{P}^{(t)} \boldsymbol{X}_{\text{train}} \in \mathbb{R}^{N \times N}$. The hidden states in the first layer are then updated by

$$\boldsymbol{Z}_{\text{train}}^{(t+1)} = \boldsymbol{Z}_{\text{train}}^{(t)} - \eta \frac{\partial L^{(t)}}{\partial \boldsymbol{Z}_{\text{train}}^{(t)}} \boldsymbol{G}_P^{(t)}. \tag{3}$$

Building on this, Theorem 2.1.1 of Wadia et al. (2021) extends to the following form.

**Assumption 1.P.** *For each $t \geq 0$, there exists a time-dependent measurable function $\Phi_t$ such that*

$$\boldsymbol{P}^{(t)} = \Phi_t(\boldsymbol{X}_{\text{train}}).$$

**Assumption 1.Q.** *For each $t \geq 0$, there exists a time-dependent measurable function $\Psi_t$ such that*

$$\boldsymbol{Q}^{(t)} = \Psi_t\big(\boldsymbol{Z}^{(t)}, \boldsymbol{\theta}_{2:L}^{(t)}, \{\boldsymbol{G}_P^{(s)}\}_{s=0}^{t-1}, \boldsymbol{Y}_{\text{train}}\big).$$

Assumptions 1.P and 1.Q state that $\boldsymbol{P}^{(t)}$ and $\boldsymbol{Q}^{(t)}$ depend only on their displayed arguments.

**Theorem 1** (Extension of Theorem 2.1.1 of Wadia et al. (2021))**.** *Assume that the initial preconditioner $\boldsymbol{P}^{(0)}$ is an arbitrary positive semi-definite matrix initialized independently of $\boldsymbol{W}_1^{(0)}$, and that the first layer is initialized in a $\boldsymbol{P}^{(0)}$-isotropic manner. Then, under Assumptions 1.P and 1.Q, for all $t \geq 1$, in terms of the mutual information $I(\cdot; \cdot)$,*

$$I\big((\boldsymbol{Z}_{\text{train}}^{(t)}, \boldsymbol{\theta}_{2:L}^{(t)}); \boldsymbol{X}_{\text{train}} \,\big|\, \{\boldsymbol{G}_P^{(s)}\}_{s=0}^{t-1}, \boldsymbol{Y}_{\text{train}}\big) = 0. \tag{4}$$

The proof is shown in Appendix A. Theorem 1 claims that, at any step $t \geq 1$, no information about $\boldsymbol{X}_{\text{train}}$ beyond what is contained in the Gram history $\{\boldsymbol{G}_P^{(s)}\}_{s=0}^{t-1}$ and the labels $\boldsymbol{Y}_{\text{train}}$ is present in the training state $(\boldsymbol{Z}_{\text{train}}^{(t)}, \boldsymbol{\theta}_{2:L}^{(t)})$.

We next consider inference on a generic test sample $(\boldsymbol{x}, \boldsymbol{y}) \sim \mu$. For each step $t$, define the cross-Gram vector $\boldsymbol{c}_P^{(t)} := \boldsymbol{X}_{\text{train}}^{\top} \boldsymbol{P}^{(t)} \boldsymbol{x}$. With $\boldsymbol{z}^{(t)} := \boldsymbol{W}_1^{(t)\top} \boldsymbol{x}$, the first-layer hidden state on $\boldsymbol{x}$ evolves as

$$\boldsymbol{z}^{(t+1)} = \boldsymbol{z}^{(t)} - \eta \frac{\partial L^{(t)}}{\partial \boldsymbol{Z}_{\text{train}}^{(t)}} \boldsymbol{c}_P^{(t)}. \tag{5}$$

Analogously to Theorem 1, Theorem 2.2.1 of Wadia et al. (2021) admits the following extension.

**Theorem 2** (Extension of Theorem 2.2.1 of Wadia et al. (2021))**.** *Assume that the initial preconditioner $\boldsymbol{P}^{(0)}$ is an arbitrary positive semi-definite matrix initialized independently of $\boldsymbol{W}_1^{(0)}$, and that the first layer is initialized in a $\boldsymbol{P}^{(0)}$-isotropic manner. Then, for all $t \geq 1$, in terms of the mutual information $I(\cdot; \cdot)$,*

$$I\big((\boldsymbol{z}^{(t)}, \boldsymbol{\theta}_{2:L}^{(t)}); \boldsymbol{X}_{\text{train}} \,\big|\, \{\boldsymbol{G}_P^{(s)}\}_{s=0}^{t-1}, \{\boldsymbol{c}_P^{(s)}\}_{s=0}^{t-1}, \boldsymbol{Y}_{\text{train}}\big) = 0. \tag{6}$$

The proof is shown in Appendix B. Theorem 2 shows that, at step $t \geq 1$, the test-time prediction $f^{(t)}(\boldsymbol{x}) = g_{2:L}(\boldsymbol{z}^{(t)}; \boldsymbol{\theta}_{2:L}^{(t)})$ depends on the data only through $\{\boldsymbol{G}_P^{(s)}\}_{s=0}^{t-1}, \{\boldsymbol{c}_P^{(s)}\}_{s=0}^{t-1}$, and $\boldsymbol{Y}_{\text{train}}$.

**Implication 1.** The preconditioner $\boldsymbol{P}^{(t)}$ establishes the geometry under which input samples are compared, inducing the inner product $\langle \boldsymbol{x}, \boldsymbol{x}' \rangle_{\boldsymbol{P}^{(t)}} := \boldsymbol{x}^{\top} \boldsymbol{P}^{(t)} \boldsymbol{x}'$ and the associated Mahalanobis norm $\|\boldsymbol{x} - \boldsymbol{x}'\|_{\boldsymbol{P}^{(t)}} := \sqrt{\langle \boldsymbol{x} - \boldsymbol{x}', \boldsymbol{x} - \boldsymbol{x}' \rangle_{\boldsymbol{P}^{(t)}}}$. As a result, the model's learned representations and generalization performance are determined solely by the input similarity geometry defined by $\boldsymbol{P}^{(t)}$—with the components emphasized by $\boldsymbol{P}^{(t)}$ becoming dominant and the components it suppresses being downweighted—and by the training labels $\boldsymbol{Y}_{\text{train}}$.

### 3.2 PRECONDITIONING BASED ON COVARIANCE EIGENDECOMPOSITION

We instantiate the preconditioner by $\boldsymbol{P} := \boldsymbol{\Sigma}_X^p$ with $\boldsymbol{\Sigma}_X := \boldsymbol{X}_{\text{train}} \boldsymbol{X}_{\text{train}}^{\top}$ and $p \in \mathbb{R}$. Since $\boldsymbol{P}$ is time-independent in this section, we simply write $\boldsymbol{P}, \boldsymbol{G}_P := \boldsymbol{X}_{\text{train}}^{\top} \boldsymbol{P} \boldsymbol{X}_{\text{train}}$, and $\boldsymbol{c}_P := \boldsymbol{X}_{\text{train}}^{\top} \boldsymbol{P} \boldsymbol{x}$.

This instantiation is reasonable and is frequently employed in the analysis of preconditioning (Wadia et al., 2021; Amari et al., 2021; Zhang et al., 2025). For least-squares regression, the Hessian is proportional to the empirical covariance, while in logistic regression and generalized linear models (GLMs), the Hessian coincides exactly with a weighted covariance. Moreover, for our model with a fully connected first layer, the Hessian with respect to each neuron in the first layer also takes the form of a weighted covariance matrix, as shown in Appendix C.

Assume $d_x \leq N$ and let the thin SVD of $\boldsymbol{X}_{\text{train}}$ be $\boldsymbol{X}_{\text{train}} = \boldsymbol{U}_{d_x} \boldsymbol{S}_{d_x} \boldsymbol{V}_{d_x}^{\top}$, where $\boldsymbol{U}_{d_x} \in \mathbb{R}^{d_x \times d_x}$, $\boldsymbol{V}_{d_x} \in \mathbb{R}^{N \times d_x}$, $\boldsymbol{S}_{d_x} = \text{diag}(s_1, \ldots, s_{d_x})$, and $r = \text{rank}(\boldsymbol{X}_{\text{train}})$. Then

$$\boldsymbol{\Sigma}_X = \boldsymbol{X}_{\text{train}} \boldsymbol{X}_{\text{train}}^{\top} = \boldsymbol{U}_{d_x} \boldsymbol{S}_{d_x}^2 \boldsymbol{U}_{d_x}^{\top}, \qquad \boldsymbol{P} = \boldsymbol{\Sigma}_X^p = \boldsymbol{U}_{d_x} \boldsymbol{S}_{d_x}^{2p} \boldsymbol{U}_{d_x}^{\top}.$$

Consequently, $\boldsymbol{G}_P$ admits the decomposition

$$\boldsymbol{G}_P \;=\; \boldsymbol{X}_{\text{train}}^{\top}\boldsymbol{P}\,\boldsymbol{X}_{\text{train}} \;=\; \boldsymbol{V}_{d_x}\boldsymbol{S}_{d_x}^{2(p+1)}\boldsymbol{V}_{d_x}^{\top} \;=\; \sum_{r=1}^{d_x} s_r^{2(p+1)}\,\boldsymbol{v}_r\boldsymbol{v}_r^{\top}. \qquad (7)$$

For an input $\boldsymbol{x} \in \mathbb{R}^{d_x}$, express $\boldsymbol{x}$ in the left–singular basis with one extra $\boldsymbol{S}_{d_x}$ factor as $\boldsymbol{x} = \boldsymbol{U}_{d_x}\boldsymbol{S}_{d_x}\,\boldsymbol{\beta}$ where $\boldsymbol{\beta} \in \mathbb{R}^{d_x}$. Then $\boldsymbol{c}_P$ becomes

$$\boldsymbol{c}_P \;=\; \boldsymbol{X}_{\text{train}}^{\top}\boldsymbol{P}\,\boldsymbol{x} \;=\; \boldsymbol{V}_{d_x}\boldsymbol{S}_{d_x}^{2(p+1)}\boldsymbol{\beta} \;=\; \sum_{r=1}^{d_x} s_r^{2(p+1)}\,\beta_r\,\boldsymbol{v}_r. \qquad (8)$$

**Implication 2.** Combining Eqs. 7 and 8 with Eqs. 4 and 6, we observe that feature learning and generalization are determined solely by Gram matrix induced by the $p$–dependent similarity geometry together with the training labels $\boldsymbol{Y}_{\text{train}}$. Within this geometry, increasing $p$ amplifies the contribution of high variance directions, whereas decreasing $p$ shifts emphasis toward low variance components of input data.

### 3.3 Preconditioning and Label Alignment for Generalization

Furthermore, within the single-index teacher framework, we make explicit the relationship between the labels and the input spectrum.

$$f^{\star}(\boldsymbol{x}) \;=\; h^{\star}\!\left(\sum_{r=1}^{d}\frac{\alpha_r}{s_r}\,\boldsymbol{u}_r^{\top}\boldsymbol{x}\right) + \boldsymbol{\epsilon} = h^{\star}\!\left(\sum_{r=1}^{d}\alpha_r\,\beta_r\right) + \boldsymbol{\epsilon}, \qquad (9)$$

where $h^{\star}:\mathbb{R}\to\mathbb{R}^{d_y}$ and $\boldsymbol{\epsilon}$ is zero-mean noise independent of $\boldsymbol{x}$. The coefficient vector $\boldsymbol{\alpha} \in \mathbb{R}^{r}$ has entries $\alpha_r$, each specifying how strongly the label depends on the corresponding principal direction.

**Implication 3.** The labels are parameterized by $\boldsymbol{\alpha}^{\top}\boldsymbol{\beta}$ and by the noise term $\boldsymbol{\epsilon}$. Building upon the discussion in Secs. 3.1 and 3.2, we conclude that the model's feature learning and generalization are determined by the Gram matrix induced by the $p$–dependent similarity geometry and the teacher's alignment with the input spectrum as captured by $\boldsymbol{\alpha}^{\top}\boldsymbol{\beta}$ (and the label noise $\boldsymbol{\epsilon}$). **Holding the input spectrum fixed, learning is governed by how $p$ drives the learner to emphasize high variance or low variance components, which components the teacher relies on to produce labels, and the level of label noise.** While we set the teacher model as single-index here, the mechanism remains unchanged when extended to a multi-index teacher: learning is facilitated as long as the spectral region emphasized by the preconditioner overlaps with the region where the teacher signal resides. In this extension, the alignment target simply shifts from a single vector to a subspace.

Motivated by this implication, the subsequent sections examine how the relationship between $p$ and the teacher's alignment with the input spectrum shapes generalization performance from several perspectives.

## 4 Multifaceted Analysis and Empirical Validation

### 4.1 On Robustness to Noise

We first empirically examine the relationship between preconditioning and vanilla generalization performance.

We construct synthetic datasets under the teacher-student setup in Eq. 9, where the teacher activation function is given by $h^{\star}(z) = \log(1 + \exp(10z))$. We specify the eigenvalue matrix $\boldsymbol{S}_{d_x}^2$ and the teacher coefficient vector $\boldsymbol{\alpha}$ as in Table 1. In Case High, only the unique largest eigenvalue corresponds to the informative signal, whereas in Case Low, only the unique smallest eigenvalue carries the signal. In both cases, we set the eigenvector matrix $\boldsymbol{U}$ to be the $d_x$-dimensional identity matrix $\boldsymbol{I}_{d_x}$, fix the dimensionality as $d_x = 10$, $d_y = 1$, and set $\lambda = 10$. Gaussian noise is added to the labels, with $\epsilon \sim \mathcal{N}(0, \sigma^2)$, where the noise standard deviation $\sigma$ is chosen to achieve a target signal-to-noise ratio (SNR). We vary SNR across $\{5, 4, 3, 2, 1\}$.

Table 1: Teacher alignment settings

| Case | $\boldsymbol{S}_{d_x}^2$ | $\boldsymbol{\alpha}$ |
|------|------|------|
| High | $\mathrm{diag}(\lambda, \lambda^{-1}, \ldots, \lambda^{-1})$ | $\boldsymbol{e}_1$ |
| Low | $\mathrm{diag}(\lambda, \ldots, \lambda, \lambda^{-1})$ | $\boldsymbol{e}_d$ |

*Note:* $\boldsymbol{e}_i$ denotes the standard basis vector whose $i$-th entry is 1 and all other entries are 0.

The student model is a two-layer MLP with ReLU activation and hidden dimension $d_h = 256$. We use only 200 training samples so that the effect of label noise becomes more pronounced. The model is trained with full-batch training for 10,000 steps, using a learning rate of $1 \times 10^{-2}$ and weight decay of $1 \times 10^{-6}$. We report the mean and standard deviation over 10 different random seeds.

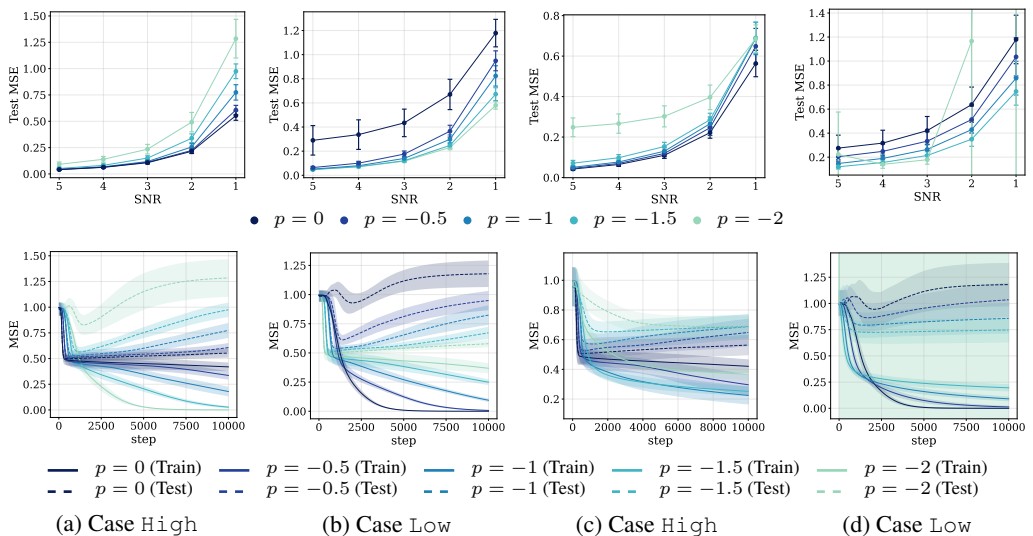

Figure 1: **The relationship between robustness to noise and preconditioning.** (a) and (b) show the results when preconditioning is performed using the exact covariance matrix, for Case High and Low, respectively. The top row presents the final test MSE for each SNR value, while the bottom row shows the train/test MSE trajectories for SNR=1. In Case High, larger $p$ values more effectively prevent overfitting, whereas in Case Low, the opposite trend is observed. (c) and (d) display the results for preconditioning with AdaHessian. Except for the numerically unstable case of $p = -2$, the same trends as in (a) and (b) are observed here as well.

### 4.1.1 PRECONDITIONING WITH THE COVARIANCE MATRIX

We first consider the case where the exact covariance matrix $\mathbf{\Sigma}_X$ is applied only to the gradients of the first-layer neurons. For the second layer, we use GD without any preconditioning. The left two columns of Fig. 1 summarize the results for both cases. The line color represents the preconditioning exponent $p$, ranging from blue ($p = 0$) to green ($p = -2$).

In the top row, we plot the relationship between SNR and the final test MSE. Across all SNR levels, we consistently observe that in Case High larger $p$ yields lower test MSE, whereas in Case Low smaller $p$ yields lower test MSE. This effect becomes more pronounced as label noise dominates.

In the bottom row, we plot the train/test MSE trajectories when SNR= 1. The trend with respect to $p$ clearly reverses between the two cases: in Case High, larger $p$ is more robust to overfitting, while in Case Low, smaller $p$ is more robust.

### 4.1.2 PRECONDITIONING WITH THE HESSIAN

Next, we examine whether preconditioning with the Hessian exhibits behavior consistent with the exact covariance-based preconditioning discussed in the previous subsection. Here, we apply preconditioning in a neuron-wise manner by using a diagonal approximation of the full parameter Hessian. Because the eigenvector matrix $\boldsymbol{U}$ is set to the identity, the covariance matrix becomes diagonal. Therefore, if the Hessian is regarded as its approximation, we expect that the diagonal approximation of the Hessian and the neuron-wise block-diagonal approximation in the first layer will exhibit equivalent behavior. In our experiments, we adopt AdaHessian (Yao et al., 2021) as an optimizer that employs the diagonal approximation of the Hessian, and we investigate how varying the power $p$ applied to its diagonal entries affects the learning dynamics.

The results are shown in the right two columns of Fig. 1. Mirroring the left two columns, the top row plots SNR against the final test MSE, and the bottom row shows the train/test MSE trajectories at SNR= 1 for each value of $p$. Except for $p = -2$ in Case Low, where training became numerically unstable and diverged, we observe the same qualitative trends as with exact covariance preconditioning: in Case High, larger $p$ consistently yields lower test MSE, whereas in Case Low, smaller $p$ yields lower test MSE, with the effect strengthening as SNR decreases.

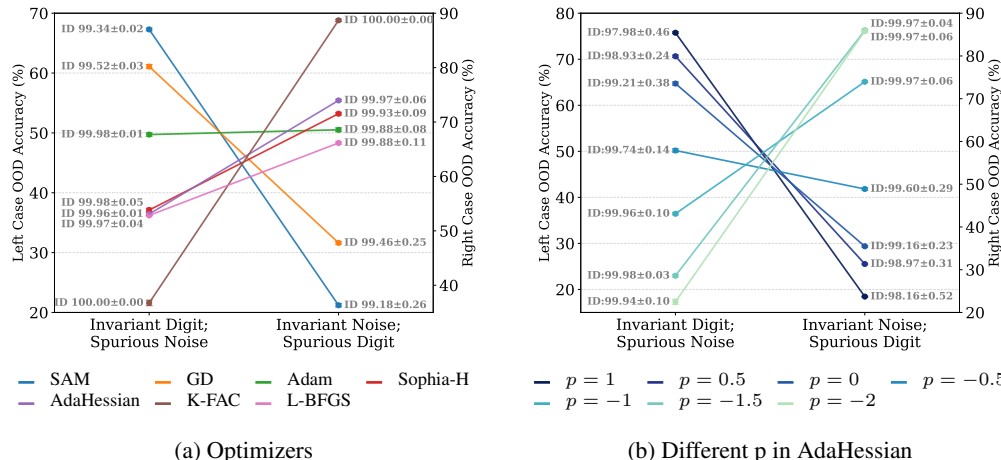

(a) Optimizers    (b) Different p in AdaHessian

Figure 2: **The relationship between OOD generalization and preconditioning. (a)** Comparison across optimizers (SAM, GD, Adam, Sophia-H, AdaHessian, K-FAC, L-BFGS). **(b)** AdaHessian with different powers applied to the diagonal Hessian entries. In each panel, the left and right columns correspond to two settings (left: invariant digit with spurious noise; right: invariant noise with a spurious digit). Although ID Val accuracy (gray numbers) is near ceiling for all methods, OOD accuracy varies substantially, and the optimizer ranking reverses between the two settings. Sweeping the power in (b) reproduces the same reversal, indicating that preconditioning steers learning toward different covariance eigen-directions; OOD performance improves when this implicit emphasis aligns with invariant features.

**Overall, in terms of robustness to noise, our results indicate that feature learning becomes more sensitive to the components emphasized by $p$, and that performance improves when $p$ is chosen to align with the teacher's spectral emphasis $\alpha$.**

**Relation to prior work.** Wadia et al. (2021) argue that whitening or its second–order proxies can harm generalization in their setting by reducing the information carried in the Gram. Our analysis suggests a more spectrum-dependent picture: the *lost* information is primarily variance–strength, which is beneficial when the teacher aligns with high–variance directions but can be *harmful* when the signal lives in low–variance directions—precisely the regime where flattening (e.g., $p = -1$) removes a misleading bias and can improve generalization. Furthermore, our results are consistent with those of Amari et al. (2021). They decomposed the test loss into bias and variance components, and demonstrated in overparameterized linear models that preconditioning aligned with the teacher is optimal for the bias term, which is the only component affected by the teacher's spectral bias. Our findings provide strong evidence that their argument can be extended to general neural networks.

## 4.2 ON OUT-OF-DISTRIBUTION GENERALIZATION

The trends observed for the choice of $p$ and generalization performance should not only be interpreted in terms of in-distribution (ID) generalization, but also be discussed in the context of OOD generalization. In particular, under correlation shift, the training inputs contain both *invariant features* and *spurious features*, where the former remain predictive under the test distribution while the latter become invalid or even harmful for generalization (Arjovsky et al., 2019; Sagawa et al., 2020). In this setting, unlike in Sec. 4.1, where the key factor was the alignment of the teacher signal with certain eigen-components of $\Sigma_X$, here the placement of spurious features along the eigen-directions also plays a decisive role in determining generalization. For example, if spurious features are aligned with low-eigenvalue components in Case `High`, then smaller values of $p$ make the learner more vulnerable to being misled. Conversely, if spurious features align with high-eigenvalue components in Case `Low`, larger values of $p$ are more detrimental.

We conduct experiments on MNIST with spurious noise added to the data. Specifically, for each of the 10 digit classes, we add class-specific Gaussian noise with a small standard deviation. We here regard the digit information as aligned with high-variance components and the added noise as

aligned with low-variance components. Although the noise is isotropic and is added equally in all directions, its relative effect becomes more significant along directions with smaller eigenvalues. At test time, we manipulate the features by either flipping the noise or flipping the digit relative to the label, thereby controlling which feature is invariant and which is spurious. In the former case, the digit acts as the invariant feature and the noise as the spurious feature, while in the latter case the roles are reversed. The model architecture is the same as in Sec. 4.1.

### 4.2.1 COMPARISON ACROSS OPTIMIZERS

We begin by comparing a range of optimizers, including SAM, GD, Adam, AdaHessian, Sophia-H, K-FAC, and L-BFGS. In our framework, SAM is treated as corresponding to $p = 1$, GD to $p = 0$, Adam to $p = -0.5$, and the remaining second-order methods as approximations of $p = -1$. K-FAC performs preconditioning by applying distinct matrices to both the input and output sides of each layer, whereas in our analysis the matrix $\boldsymbol{P}$ corresponds to the input-side factor. Hyperparameters are selected by grid search based on accuracy on ID validation (Val) accuracy.

Fig. 2a reports the results. We plot the case where the noise is flipped as the left column of points and the case where the digit is flipped as the right column. Each color denotes an optimizer; the left (right) y-axis corresponds to the OOD accuracy of the left (right) column. For reference, we report the ID Val accuracy next to each point in gray. Despite near-ceiling ID Val accuracy for all methods, OOD accuracy differs substantially across optimizers and, crucially, the ranking flips between the two settings. When the noise is spurious (left column), optimizers that emphasize higher-variance directions perform better (SAM > GD > second-order optimizers). Conversely, when the digit is spurious (right column), second-order optimizers dominate (second-order optimizers > GD > SAM). Notably, within the family of second-order methods, substantial variation remains: their relative ranking also tends to flip. Some, such as Adam, exhibit behavior more closely aligned with GD and SAM, whereas others, such as K-FAC, diverge more significantly. The ranking within second-order optimizers is discussed in Appendix D.

### 4.2.2 COMPARISON ACROSS POWERS OF THE HESSIAN

Following Sec. 4.1.2, we vary the exponent $p$ applied to the AdaHessian preconditioner and independently tune hyperparameters for each $p$.

The results in Fig. 2b mirror the optimizer-level comparison in Fig. 2a: the OOD ranking flips between the two settings, while ID validation remains near ceiling for all $p$. Thus, increasing or decreasing $p$ systematically changes which spectral components the learner relies on, reinforcing the interpretation that Hessian-based preconditioning acts as a controllable spectral bias whose optimal value depends on the alignment between invariant/spurious features and the eigenstructure of $\boldsymbol{\Sigma}_X$.

**As in ID generalization, we find that OOD performance hinges on the preconditioner emphasizing the same spectral directions as the teacher's signal. More concretely, a preconditioner that emphasizes invariant feature components while suppressing spurious components improves OOD generalization.**

### 4.3 ON KNOWLEDGE FORWARD TRANSFER

In transfer and continual learning, an important objective is to exploit knowledge from prior tasks to enhance performance on later tasks, a process commonly referred to as *forward transfer*. The extent to which knowledge can be transferred largely depends on the spectral components that a model has learned to depend on in the source task. Since future task information is generally unavailable during pretraining, a task-agnostic approach is to learn features in a uniform manner across covariance eigen-directions. In what follows, we investigate which choices of preconditioner in the source task improve transferability to subsequent tasks, by controlling the spectral emphasis through the $p$.

We adapt the synthetic setup from Sec. 4.1. Each case (High, Low) now comprises two tasks, Task1 and Task2. For symmetry between tasks, we now set $\boldsymbol{S}_{d_x}^2 = \lambda \boldsymbol{I}_{d/2} \oplus \lambda^{-1} \boldsymbol{I}_{d/2}$ with $d$ even. In Task1, the teacher puts its signal on one extreme eigen-direction (largest for High, smallest for Low), so larger (smaller) $p$ should help in High (Low). In Task2, the Task1 signal is made non-informative and moved to one opposite extreme direction. We first train in Task1 under varying $p$. For Task2, we initialize from the model trained in Task1, freeze the first layer, and optimize only the second layer.

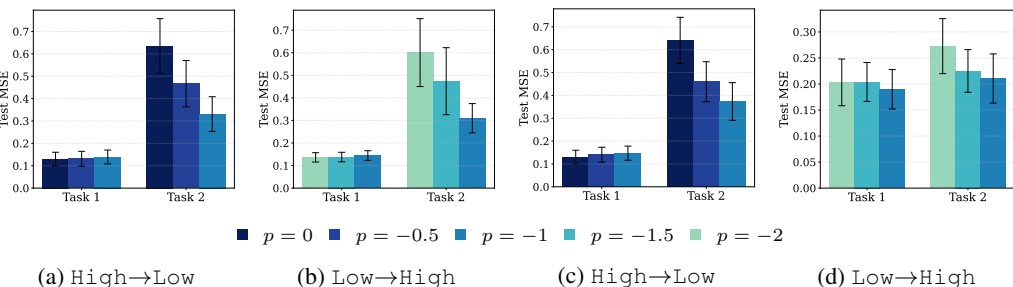

(a) High→Low      (b) Low→High      (c) High→Low      (d) Low→High

Figure 3: **The relationship between knowledge transferability and preconditioning.** (a), (b) show the case where exact covariance preconditioning with different exponents $p$ is applied in Task1, and (c), (d) show the case where $p$ is swept using AdaHessian. For each case (High, Low), the test MSE of Task1/Task2 is shown. In Task1, generalization improves as $p$ increases in High, and as $p$ decreases in Low. In contrast, in Task2, where only the second layer is optimized while other components are fixed to model trained in Task1, $p = -1$ gives the best result in both cases. Similar trends are reproduced in (c), (d), indicating that the spectral bias formed in Task1 strongly influences knowledge transferability in Task2.

Since this is convex, we solve it by ridge regression in closed form; the regularization coefficient is tuned on Task2 validation MSE.

Fig. 3 summarizes the results. The left two columns vary the power $p$ for exact covariance preconditioning, and the right two columns vary $p$ for AdaHessian; within each pair, the left column is High and the right column is Low. For each case we contrast $p = -1$ against larger values ($p \in \{-0.5, 0\}$) and smaller values ($p \in \{-1.5, -2\}$). These bar plots show Task 1 and Task 2 test MSE across $p$. On Task 1, $p$ values larger (smaller) than $-1$ achieve better generalization in High (Low) in most cases, consistent with Sec. 4.1. In contrast, on Task 2 the best generalization is attained at $p = -1$ for both High and Low.

The spectral bias induced by preconditioning on the source task strongly governs forward transfer: choosing $p = -1$, which treats all covariance eigen-directions uniformly, yields features that transfer broadly, accelerating optimization and improving generalization on subsequent tasks. **While second-order optimization has recently attracted attention in large-scale pretraining primarily for its potential to reduce training cost, we argue that it is likewise warranted from the standpoint of knowledge-transfer performance.** While Zhang et al. (2024; 2025) demonstrated the effectiveness of whitening and K-FAC for feature learning in transfer learning, our framework explains these findings as a specific instance ($p = -1$) of a broader spectral control mechanism, where minimizing spectral bias maximizes the preservation of transferrable features.

## 5 LIMITATION AND FUTURE WORK

Our insights, while strongly suggestive of our claims, do not constitute a formal proof. A more rigorous discussion, grounded in the theoretical analysis of deep learning dynamics, is necessary and we leave this for future work.

We also envision that our work could ultimately lead to a framework capable of automatically determining the optimal preconditioner to satisfy a user's desired trade-offs between generalization performance and convergence speed. While it is practically impossible to fully uncover the relationship between the teacher model and the manifold structure of the input data during training, the problem may become tractable if certain prior knowledge is available. For instance, under the assumption of a monotonic teacher model, it would be possible to identify signal-bearing components by measuring the correlation between each spectral direction and the corresponding labels.

## 6 CONCLUSION

In this work, we reframed preconditioning in machine learning not merely as a tool for accelerating convergence, but as a principled mechanism for controlling which features are learned. Our central thesis is that preconditioning imposes a controllable spectral bias on feature learning by altering the

input-space similarity metric. We instantiate this bias via the $p$-th power of the input covariance matrix, where the exponent $p$ determines whether high-variance or low-variance components dominate in the induced geometry. Our expanded analysis demonstrates that aligning this spectral emphasis with the teacher's structure improves in-distribution generalization, out-of-distribution robustness, and forward knowledge transfer. These findings provide a unified, spectrum-centric perspective that helps resolve long-standing debates about optimizer choice and paves the way for task-aware optimizers that explicitly shape feature learning to achieve better generalization.

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

# A    PROOF OF THEOREM 1

In this appendix we prove Theorem 1. We begin by establishing Proposition 1, which provides the base conditional independence at initialization.

**Proposition 1.** *Let $\boldsymbol{P}^{(0)} \in \mathbb{R}^{d_x \times d_x}$ be symmetric positive semi-definite (rank r), $\boldsymbol{X}_{\text{train}} \in \mathbb{R}^{d_x \times N}$, and $\boldsymbol{W}^{(0)} \in \mathbb{R}^{d_x \times d_h}$. Assume the columns $\{\boldsymbol{w}_j\}_{j=1}^{d_h}$ of $\boldsymbol{W}^{(0)}$ are i.i.d. draws from $\mathcal{N}(0, \sigma^2 \boldsymbol{P}^{(0)})$ and independent of $\boldsymbol{X}_{\text{train}}$. Define the pre-activations $\boldsymbol{Z}^{(0)} := (\boldsymbol{W}^{(0)})^\top \boldsymbol{X}_{\text{train}} \in \mathbb{R}^{d_h \times N}$ and the $\boldsymbol{P}^{(0)}$-Gram matrix $\boldsymbol{G}_P^{(0)} := \boldsymbol{X}_{\text{train}}^\top \boldsymbol{P}^{(0)} \boldsymbol{X}_{\text{train}}$. Then*

$$I(\boldsymbol{Z}^{(0)}; \boldsymbol{X}_{\text{train}} \,|\, \boldsymbol{G}_P^{(0)}) = 0.$$

*Proof.* Let $\boldsymbol{P}^{(0)} = \boldsymbol{S}^\top \boldsymbol{S}$ be a (rank-$r$) factorization with $\boldsymbol{S} \in \mathbb{R}^{r \times d_x}$. For each column $\boldsymbol{w}_j$ of $\boldsymbol{W}^{(0)}$, there exists $\boldsymbol{u}_j \sim \mathcal{N}(0, \sigma^2 \boldsymbol{I}_r)$ such that $\boldsymbol{w}_j = \boldsymbol{S}^\top \boldsymbol{u}_j$ in distribution. Stacking the $\boldsymbol{u}_j$ as columns gives $\boldsymbol{U} \in \mathbb{R}^{r \times d_h}$ with i.i.d. columns $\boldsymbol{u}_j \sim \mathcal{N}(0, \sigma^2 \boldsymbol{I}_r)$ and

$$\boldsymbol{W}^{(0)} \overset{d}{=} \boldsymbol{S}^\top \boldsymbol{U}.$$

Define the reduced data $\boldsymbol{X}'_{\text{train}} := \boldsymbol{S}\boldsymbol{X}_{\text{train}} \in \mathbb{R}^{r \times N}$. Then

$$\boldsymbol{Z}^{(0)} = (\boldsymbol{W}^{(0)})^\top \boldsymbol{X}_{\text{train}} = (\boldsymbol{S}^\top \boldsymbol{U})^\top \boldsymbol{X}_{\text{train}} = \boldsymbol{U}^\top (\boldsymbol{S}\boldsymbol{X}_{\text{train}}) = \boldsymbol{U}^\top \boldsymbol{X}'_{\text{train}}.$$

Because $\boldsymbol{U}$ has isotropic Gaussian columns and is independent of $\boldsymbol{X}'_{\text{train}}$, the isotropic case (Appendix A / Wadia et al. (2021)) gives

$$I\Big(\boldsymbol{Z}^{(0)}; \boldsymbol{X}'_{\text{train}} \,\Big|\, (\boldsymbol{X}'_{\text{train}})^\top \boldsymbol{X}'_{\text{train}}\Big) = 0.$$

Noting $(\boldsymbol{X}'_{\text{train}})^\top \boldsymbol{X}'_{\text{train}} = \boldsymbol{X}_{\text{train}}^\top \boldsymbol{S}^\top \boldsymbol{S} \boldsymbol{X}_{\text{train}} = \boldsymbol{X}_{\text{train}}^\top \boldsymbol{P}^{(0)} \boldsymbol{X}_{\text{train}} = \boldsymbol{G}_P^{(0)}$, we have

$$I\Big(\boldsymbol{Z}^{(0)}; \boldsymbol{X}'_{\text{train}} \,\Big|\, \boldsymbol{G}_P^{(0)}\Big) = 0.$$

Finally, $\boldsymbol{Z}^{(0)} = \boldsymbol{U}^\top \boldsymbol{X}'_{\text{train}}$ implies the Markov chain $\boldsymbol{X}_{\text{train}} \to \boldsymbol{X}'_{\text{train}} \to \boldsymbol{Z}^{(0)}$ (since $\boldsymbol{U} \perp \boldsymbol{X}_{\text{train}}$), and hence by the conditional data-processing inequality

$$I\Big(\boldsymbol{Z}^{(0)}; \boldsymbol{X}_{\text{train}} \,\Big|\, \boldsymbol{G}_P^{(0)}\Big) \leq I\Big(\boldsymbol{Z}^{(0)}; \boldsymbol{X}'_{\text{train}} \,\Big|\, \boldsymbol{G}_P^{(0)}\Big) = 0,$$

as claimed. $\square$

We now prove Theorem 1 using the following lemmas.

**Lemma 1.** *Let $A, X, C, Y$ be random elements. and let $B = \psi(A, C, Y)$ be a measurable function of $(A, C, Y)$. If $A \perp X \mid C$ and $Y \perp A \mid (X, C)$, then $B \perp X \mid (C, Y)$.*

**Lemma 2.** *Let $A, X, C$ be random elements and let $D = \phi(X, C)$ be a measurable function of $(X, C)$. If $A \perp X \mid C$, then $A \perp X \mid (C, D)$.*

By Proposition 1, $\boldsymbol{Z}^{(0)} \perp \boldsymbol{X}_{\text{train}} \mid \boldsymbol{G}_P^{(0)}$. Since $\boldsymbol{\theta}_{2:L}^{(0)}$ is initialized independently of $(\boldsymbol{X}_{\text{train}}, \boldsymbol{P}^{(0)})$, we also have $(\boldsymbol{Z}^{(0)}, \boldsymbol{\theta}_{2:L}^{(0)}) \perp \boldsymbol{X}_{\text{train}} \mid \boldsymbol{G}_P^{(0)}$. Moreover, by the data-generating assumption, $\boldsymbol{Y}_{\text{train}} \perp (\boldsymbol{Z}^{(0)}, \boldsymbol{\theta}_{2:L}^{(0)}) \mid (\boldsymbol{X}_{\text{train}}, \boldsymbol{G}_P^{(0)})$.

By Eq. 3, the update rule of $\boldsymbol{\theta}_{2:L}$ and Assumption 1.Q, the updates at $t = 0$ are measurable functions of $(\boldsymbol{Z}^{(0)}, \boldsymbol{\theta}_{2:L}^{(0)}, \boldsymbol{Y}_{\text{train}}, \boldsymbol{G}_P^{(0)})$, i.e.,

$$(\boldsymbol{Z}^{(1)}, \boldsymbol{\theta}_{2:L}^{(1)}) = \psi_0\big(\boldsymbol{Z}^{(0)}, \boldsymbol{\theta}_{2:L}^{(0)}, \boldsymbol{Y}_{\text{train}}, \boldsymbol{G}_P^{(0)}\big)$$

for some measurable $\psi_0$. Applying Lemma 1 with $A = (\boldsymbol{Z}^{(0)}, \boldsymbol{\theta}_{2:L}^{(0)})$, $X = \boldsymbol{X}_{\text{train}}$, $C = \boldsymbol{G}_P^{(0)}$, $Y = \boldsymbol{Y}_{\text{train}}$, we obtain

$$(\boldsymbol{Z}^{(1)}, \boldsymbol{\theta}_{2:L}^{(1)}) \perp \boldsymbol{X}_{\text{train}} \,\big|\, (\boldsymbol{G}_P^{(0)}, \boldsymbol{Y}_{\text{train}}),$$

which is Eq. 4 for $t = 1$.

Assume for some $t \geq 1$ that

$$(\boldsymbol{Z}^{(t)}, \boldsymbol{\theta}_{2:L}^{(t)}) \perp \boldsymbol{X}_{\text{train}} \mid (\{\boldsymbol{G}_P^{(s)}\}_{s=0}^{t-1}, \boldsymbol{Y}_{\text{train}}). \tag{10}$$

By Assumption 1.P, $\boldsymbol{G}_P^{(t)} = \phi_t(\boldsymbol{X}_{\text{train}})$ depends only on $\boldsymbol{X}_{\text{train}}$. Applying Lemma 2 to Eq. 10 with $A = (\boldsymbol{Z}^{(t)}, \boldsymbol{\theta}_{2:L}^{(t)}), C = (\{\boldsymbol{G}_P^{(s)}\}_{s=0}^{t-1}, \boldsymbol{Y}_{\text{train}}), D = \boldsymbol{G}_P^{(t)}$ yields

$$(\boldsymbol{Z}^{(t)}, \boldsymbol{\theta}_{2:L}^{(t)}) \perp \boldsymbol{X}_{\text{train}} \mid (\{\boldsymbol{G}_P^{(s)}\}_{s=0}^{t}, \boldsymbol{Y}_{\text{train}}). \tag{11}$$

By Eq. 3, the update rule of $\boldsymbol{\theta}_{2:L}$ and Assumption 1.Q, $(\boldsymbol{Z}^{(t+1)}, \boldsymbol{\theta}_{2:L}^{(t+1)})$ is a measurable function of $(\boldsymbol{Z}^{(t)}, \boldsymbol{\theta}_{2:L}^{(t)}, \boldsymbol{Y}_{\text{train}}, \boldsymbol{G}_P^{(t)})$. and applying Eq. 11 and Lemma 1 with $C = (\{\boldsymbol{G}_P^{(s)}\}_{s=0}^{t}, \boldsymbol{Y}_{\text{train}})$ gives

$$(\boldsymbol{Z}^{(t+1)}, \boldsymbol{\theta}_{2:L}^{(t+1)}) \perp \boldsymbol{X}_{\text{train}} \mid (\{\boldsymbol{G}_P^{(s)}\}_{s=0}^{t}, \boldsymbol{Y}_{\text{train}}),$$

i.e.,

$$I\big((\boldsymbol{Z}^{(t+1)}, \boldsymbol{\theta}_{2:L}^{(t+1)}); \boldsymbol{X}_{\text{train}} \mid \{\boldsymbol{G}_P^{(s)}\}_{s=0}^{t}, \boldsymbol{Y}_{\text{train}}\big) = 0.$$

This is Eq. 4 with $t$ replaced by $t+1$, completing the induction.

□

## B  PROOF OF THEOREM 2

We prove Theorem 2 by reusing Proposition 1 and the inductive scheme of Theorem 1, together with the following lemma.

**Lemma 3.** *Let $A, X, C, R$ be random elements with $R \perp (A, X, C)$ and let $D = \phi(X, C, R)$ be measurable. If $A \perp X \mid C$, then $A \perp X \mid (C, D)$.*

By applying the same reasoning as in Appendix A, we obtain $(\boldsymbol{Z}_{\text{train}}^{(0)}, \boldsymbol{\theta}_{2:L}^{(0)}) \perp \boldsymbol{X}_{\text{train}} \mid \boldsymbol{G}_P^{(0)}$. Moreover $\boldsymbol{z}^{(0)} = \boldsymbol{W}_1^{(0)\top} \boldsymbol{x}$ depends only on $(\boldsymbol{W}_1^{(0)}, \boldsymbol{x})$ which are independent of $\boldsymbol{X}_{\text{train}}$, so $(\boldsymbol{z}^{(0)}, \boldsymbol{Z}_{\text{train}}^{(0)}, \boldsymbol{\theta}_{2:L}^{(0)}) \perp \boldsymbol{X}_{\text{train}} \mid \boldsymbol{G}_P^{(0)}$. Since $\boldsymbol{c}_P^{(0)} = \boldsymbol{X}_{\text{train}}^{\top} \boldsymbol{P}^{(0)} \boldsymbol{x}$ is a measurable function of $(\boldsymbol{X}_{\text{train}}, \boldsymbol{G}_P^{(0)}, \boldsymbol{x})$ and $\boldsymbol{x} \perp (\boldsymbol{X}_{\text{train}}, \boldsymbol{G}_P^{(0)}, \boldsymbol{z}^{(0)}, \boldsymbol{Z}_{\text{train}}^{(0)}, \boldsymbol{\theta}_{2:L}^{(0)})$, Lemma 3 with $A = (\boldsymbol{z}^{(0)}, \boldsymbol{Z}_{\text{train}}^{(0)}, \boldsymbol{\theta}_{2:L}^{(0)}), C = \boldsymbol{G}_P^{(0)}, R = \boldsymbol{x}, D = \boldsymbol{c}_P^{(0)}$ gives

$$(\boldsymbol{z}^{(0)}, \boldsymbol{Z}_{\text{train}}^{(0)}, \boldsymbol{\theta}_{2:L}^{(0)}) \perp \boldsymbol{X}_{\text{train}} \mid (\boldsymbol{G}_P^{(0)}, \boldsymbol{c}_P^{(0)}).$$

Using the update equations (Eqs. 3, 5) and Assumption 1.Q, there exists a measurable map $\psi_0$ such that

$$(\boldsymbol{z}^{(1)}, \boldsymbol{\theta}_{2:L}^{(1)}) = \psi_0\big(\boldsymbol{z}^{(0)}, \boldsymbol{Z}_{\text{train}}^{(0)}, \boldsymbol{\theta}_{2:L}^{(0)}, \boldsymbol{Y}_{\text{train}}, \boldsymbol{G}_P^{(0)}, \boldsymbol{c}_P^{(0)}\big).$$

By Lemma 1, we obtain

$$(\boldsymbol{z}^{(1)}, \boldsymbol{\theta}_{2:L}^{(1)}) \perp \boldsymbol{X}_{\text{train}} \mid (\boldsymbol{G}_P^{(0)}, \boldsymbol{c}_P^{(0)}, \boldsymbol{Y}_{\text{train}}),$$

which is the claim for $t = 1$.

Assume for some $t \geq 1$ that

$$(\boldsymbol{z}^{(t)}, \boldsymbol{\theta}_{2:L}^{(t)}) \perp \boldsymbol{X}_{\text{train}} \mid (\{\boldsymbol{G}_P^{(s)}\}_{s=0}^{t-1}, \{\boldsymbol{c}_P^{(s)}\}_{s=0}^{t-1}, \boldsymbol{Y}_{\text{train}}).$$

By Assumption 1.P, $\boldsymbol{G}_P^{(t)} = \phi_t(\boldsymbol{X}_{\text{train}})$ depends only on $\boldsymbol{X}_{\text{train}}$, so Lemma 2 implies

$$(\boldsymbol{z}^{(t)}, \boldsymbol{\theta}_{2:L}^{(t)}) \perp \boldsymbol{X}_{\text{train}} \mid (\{\boldsymbol{G}_P^{(s)}\}_{s=0}^{t}, \{\boldsymbol{c}_P^{(s)}\}_{s=0}^{t-1}, \boldsymbol{Y}_{\text{train}}).$$

Since $\boldsymbol{c}_P^{(t)} = \boldsymbol{X}_{\text{train}}^{\top} \boldsymbol{P}^{(t)} \boldsymbol{x}$ is a measurable function of $(\boldsymbol{X}_{\text{train}}, \{\boldsymbol{G}_P^{(s)}\}_{s=0}^{t}, \boldsymbol{x})$ and $\boldsymbol{x} \perp (\boldsymbol{X}_{\text{train}}, \{\boldsymbol{G}_P^{(s)}\}_{s=0}^{t}, \boldsymbol{z}^{(t)}, \boldsymbol{\theta}_{2:L}^{(t)})$, Lemma 3 yields

$$(\boldsymbol{z}^{(t)}, \boldsymbol{\theta}_{2:L}^{(t)}) \perp \boldsymbol{X}_{\text{train}} \mid (\{\boldsymbol{G}_P^{(s)}\}_{s=0}^{t}, \{\boldsymbol{c}_P^{(s)}\}_{s=0}^{t}, \boldsymbol{Y}).$$

By Eqs. 3, 5 and Assumption 1.Q,

$$(\boldsymbol{z}^{(t+1)}, \boldsymbol{\theta}_{2:L}^{(t+1)}) = \psi_t(\boldsymbol{z}^{(t)}, \boldsymbol{\theta}_{2:L}^{(t)}, \boldsymbol{Y}_{\text{train}}, \boldsymbol{G}_P^{(t)}, \boldsymbol{c}_P^{(t)})$$

for some measurable $\psi_t$. Applying Lemma 1 with $C = (\{\boldsymbol{G}_P^{(s)}\}_{s=0}^t, \boldsymbol{Y})$ (and noting that $\boldsymbol{c}_P^{(t)}$ is already included via $\mathcal{C}_t$) gives

$$(\boldsymbol{z}^{(t+1)}, \boldsymbol{\theta}_{2:L}^{(t+1)}) \perp \boldsymbol{X}_{\text{train}} \,\big|\, (\{\boldsymbol{G}_P^{(s)}\}_{s=0}^t, \mathcal{C}_t, \boldsymbol{Y}_{\text{train}}).$$

This is precisely the desired statement at time $t + 1$, completing the induction.

$\square$

## C   HESSIAN STRUCTURE FOR THE FIRST LAYER

The gradient with respect to the first-layer weights is

$$\frac{\partial L}{\partial \boldsymbol{W}_1} = \boldsymbol{X}_{\text{train}} \left( \frac{\partial L}{\partial \boldsymbol{Z}_{\text{train}}} \right)^\top. \tag{12}$$

Thus, for neuron $j$ with weight vector $\boldsymbol{w}_{1,j}$, we have

$$\nabla_{\boldsymbol{w}_{1,j}} L = \sum_i a_{i,j}\, \boldsymbol{x}_i, \tag{13}$$

with $a_{i,j} := \partial L/\partial z_{i,j}$. Differentiating once more gives

$$\nabla^2_{\boldsymbol{w}_{1,j}} L = \sum_i b_{i,j}\, \boldsymbol{x}_i \boldsymbol{x}_i^\top, \tag{14}$$

with $b_{i,j} := \partial \delta_{i,j}/\partial z_{i,j}$. Hence, the Hessian block for neuron $j$ is a weighted covariance matrix of the inputs, with weights $b_{i,j}$ determined by the loss curvature and backpropagated signals.

## D   MAPPING SECOND-ORDER OPTIMIZERS TO THE SPECTRAL BIAS $p$

Our theoretical framework analyzes preconditioning via the exponent $p$ in $\boldsymbol{P} = \boldsymbol{\Sigma}_{\boldsymbol{X}}^p$. Here, we interpret how the second-order optimizers align with this framework, particularly in terms of their proximity to the regime of $p = -1$.

**K-FAC.** K-FAC approximates the FIM layer-wise by decomposing it into the Kronecker product of output-gradient and input covariances:

$$\boldsymbol{F}_{\text{KFAC}} \approx \mathbb{E}\left[ \nabla_{\boldsymbol{z}} \mathcal{L} (\nabla_{\boldsymbol{z}} \mathcal{L})^\top \right] \otimes \mathbb{E}\left[ \boldsymbol{x}\boldsymbol{x}^\top \right] := \boldsymbol{G} \otimes \boldsymbol{A}. \tag{15}$$

Crucially, the input factor $\boldsymbol{A} = \mathbb{E}[\boldsymbol{x}\boldsymbol{x}^\top]$ coincides exactly with the input covariance matrix $\boldsymbol{\Sigma}_X$. Consequently, the preconditioned update rule applies the inverse of this covariance to the input side of the weight gradient:

$$\Delta \boldsymbol{W} = \boldsymbol{A}^{-1}(\nabla_{\boldsymbol{W}} \mathcal{L})\boldsymbol{G}^{-1} = \boldsymbol{\Sigma}_{\boldsymbol{X}}^{-1}(\nabla_{\boldsymbol{W}} \mathcal{L})\boldsymbol{G}^{-1}. \tag{16}$$

Since the input-side preconditioner directly determines the similarity metric affecting inference in our framework, this left-multiplication by $\boldsymbol{\Sigma}_{\boldsymbol{X}}^{-1}$ formally aligns K-FAC with the whitening regime where $p = -1$. Unlike diagonal approximations, K-FAC retains the dense structure of $\boldsymbol{\Sigma}_{\boldsymbol{X}}$, making it the implementation closest to the theoretical ideal of $p = -1$ among the evaluated methods.

**AdaHessian and Sophia-H.** These methods rely on diagonal approximations of the Hessian applied with an inverse power. While they aim for a Newton-like update, they strictly correspond to $p = -1$ only if the input features are independent (i.e., the covariance matrix is diagonal). In practice, neglecting off-diagonal components distances them from the exact spectral bias of $p = -1$ compared to K-FAC. Furthermore, they differ in their handling of small eigenvalues: AdaHessian employs soft damping, resulting in a smooth transition between $p = -1$ and $p = 0$ as eigenvalues decrease. In contrast, Sophia-H utilizes a hard clipping mechanism, creating a distinct boundary where directions with curvature below a threshold effectively revert to $p = 0$.

**L-BFGS.** L-BFGS approximates the inverse Hessian using a history of the past $m$ gradient updates. It captures curvature information ($p \approx -1$) specifically along the directions spanned by these recent updates. However, for the vast majority of the parameter space orthogonal to the history, it retains unit curvature ($p = 0$). Consequently, L-BFGS operates as a hybrid method, mixing the behaviors of $p = -1$ and $p = 0$.

**Adam.** Adam scales gradients by the inverse square root of the second moment estimate, which roughly corresponds to $p = -1/2$. This formulation places its spectral bias closer to GD than to the pure second-order methods discussed above.

**Summary.** Based on this analysis, we can order the optimizers by their proximity to the theoretical $p = -1$ regime: K-FAC is the nearest, followed by the diagonal/hybrid approximations (AdaHessian, Sophia-H, L-BFGS), with Adam being the furthest and closest to GD. The experimental results in Figure 2a are consistent with this hierarchy, demonstrating that optimizers with a stronger structural fidelity to $p = -1$ exhibit the most distinct spectral biases (e.g., excelling when low-variance features are invariant, as seen in the "Invariant Noise" setting).

# E   DERIVATION OF SAM UPDATE AS HESSIAN PRECONDITIONING

In this section, we provide a derivation showing that the SAM update rule can be approximated as a Hessian-based preconditioning of the gradient.

Recall the gradient computation in SAM, denoted as $g_{\text{SAM}}$, which involves computing the gradient at a perturbed parameter location to maximize the loss locally:

$$g_{\text{SAM}} = \nabla\mathcal{L}\left(\boldsymbol{\theta} + \frac{\rho}{||\nabla\mathcal{L}(\boldsymbol{\theta})||}\nabla\mathcal{L}(\boldsymbol{\theta})\right), \tag{17}$$

where $\rho$ is the neighborhood size. Let us define the perturbation vector $\boldsymbol{\delta}$ as:

$$\boldsymbol{\delta} = \frac{\rho}{||\nabla\mathcal{L}(\boldsymbol{\theta})||}\nabla\mathcal{L}(\boldsymbol{\theta}). \tag{18}$$

Assuming that $\rho$ is sufficiently small, we can apply a first-order Taylor expansion of the gradient $\nabla\mathcal{L}(\boldsymbol{\theta} + \boldsymbol{\delta})$ around $\boldsymbol{\theta}$. Let $\boldsymbol{H} = \nabla^2\mathcal{L}(\boldsymbol{\theta})$ denote the Hessian matrix of the loss function at $\boldsymbol{\theta}$. The approximation proceeds as follows:

$$
\begin{aligned}
g_{\text{SAM}} &\approx \nabla\mathcal{L}(\boldsymbol{\theta}) + \boldsymbol{H}\boldsymbol{\delta} \\
&= \nabla\mathcal{L}(\boldsymbol{\theta}) + \boldsymbol{H}\left(\frac{\rho}{||\nabla\mathcal{L}(\boldsymbol{\theta})||}\nabla\mathcal{L}(\boldsymbol{\theta})\right) \\
&= \nabla\mathcal{L}(\boldsymbol{\theta}) + \frac{\rho}{||\nabla\mathcal{L}(\boldsymbol{\theta})||}\boldsymbol{H}\nabla\mathcal{L}(\boldsymbol{\theta}).
\end{aligned} \tag{19}
$$

By factoring out the gradient term $\nabla\mathcal{L}(\boldsymbol{\theta})$, we obtain the final approximate form:

$$g_{\text{SAM}} \approx \left(I + \frac{\rho}{||\nabla\mathcal{L}(\boldsymbol{\theta})||}\boldsymbol{H}\right)\nabla\mathcal{L}(\boldsymbol{\theta}). \tag{20}$$

This derivation illustrates that the effective gradient used in SAM is approximately equivalent to the standard gradient preconditioned by the matrix $\left(I + \frac{\rho}{||\nabla\mathcal{L}(\boldsymbol{\theta})||}\boldsymbol{H}\right)$. Therefore, SAM corresponds to $p = 1$ in our formulation.

# F   ADDITIONAL RESULTS

## F.1   ROBUSTNESS TO NOISE

Figs. 4, 5, 6, and 7 show the training trajectories at other SNR levels corresponding to Fig. 1.

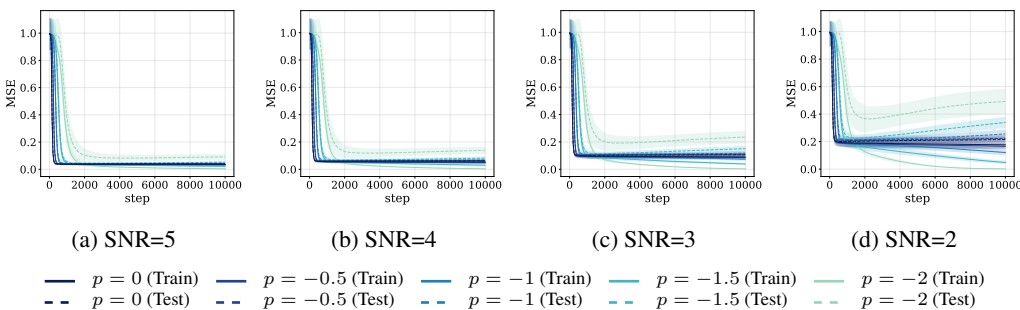

Figure 4: Training and test performance trajectories for each SNR level with the exact covariance preconditioner for different $p$ on Case `High`

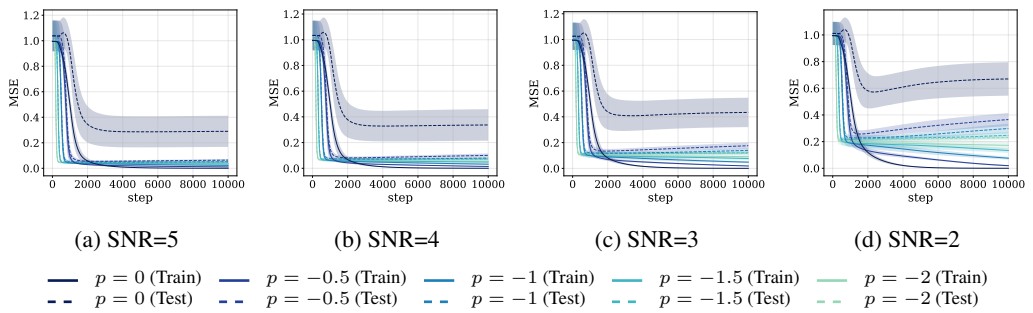

Figure 5: Training and test performance trajectories for each SNR level with the exact covariance preconditioner for different $p$ on Case `Low`

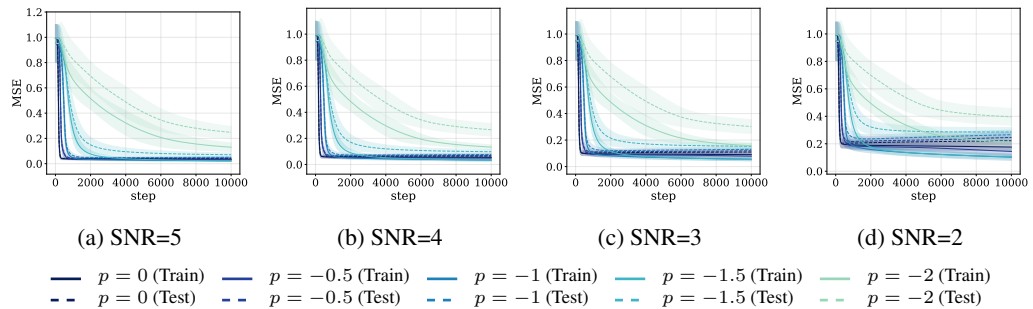

Figure 6: Training and test performance trajectories for each SNR level with the AdaHessian preconditioner for different $p$ on Case `High`

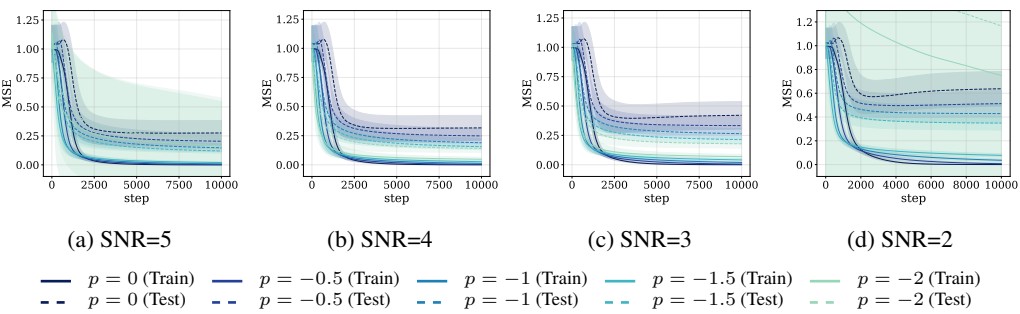

Figure 7: Training and test performance trajectories for each SNR level with the AdaHessian preconditioner for different $p$ on Case `Low`

Table 2: Tabular summary of Figures 2a and 2b

|  | SAM | GD | Adam | Sophia-H | AdaHessian | KFAC | L-BFGS |
|---|---|---|---|---|---|---|---|
| **Invariant: Digit; Spurious: Noise** | | | | | | | |
| ID Val | $99.34 \pm 0.02$ | $99.52 \pm 0.03$ | $99.98 \pm 0.01$ | $99.98 \pm 0.05$ | $99.96 \pm 0.01$ | $100.00 \pm 0.00$ | $99.97 \pm 0.04$ |
| OOD Test | $67.29 \pm 0.23$ | $61.09 \pm 0.34$ | $49.72 \pm 0.29$ | $37.11 \pm 0.31$ | $36.44 \pm 0.24$ | $21.59 \pm 0.46$ | $36.21 \pm 0.11$ |
| **Invariant: Noise; Spurious: Digit** | | | | | | | |
| ID Val | $99.18 \pm 0.26$ | $99.46 \pm 0.25$ | $99.88 \pm 0.08$ | $99.93 \pm 0.09$ | $99.97 \pm 0.06$ | $100.00 \pm 0.00$ | $99.88 \pm 0.11$ |
| OOD Test | $36.32 \pm 0.29$ | $47.79 \pm 0.26$ | $68.57 \pm 0.36$ | $71.53 \pm 0.25$ | $73.97 \pm 0.20$ | $88.69 \pm 0.22$ | $66.15 \pm 0.22$ |

(a) Comparison across optimizers.

|  | $p = 1$ | $p = 0.5$ | $p = 0$ | $p = -0.5$ | $p = -1$ | $p = -1.5$ | $p = -2$ |
|---|---|---|---|---|---|---|---|
| **Invariant: Digit; Spurious: Noise** | | | | | | | |
| ID Val | $97.98 \pm 0.46$ | $98.93 \pm 0.24$ | $99.21 \pm 0.38$ | $99.74 \pm 0.14$ | $99.96 \pm 0.10$ | $99.98 \pm 0.03$ | $99.94 \pm 0.10$ |
| OOD Test | $75.76 \pm 0.20$ | $70.65 \pm 0.26$ | $64.72 \pm 0.35$ | $50.18 \pm 0.36$ | $36.44 \pm 0.24$ | $22.98 \pm 0.28$ | $17.32 \pm 0.55$ |
| **Invariant: Noise; Spurious: Digit** | | | | | | | |
| ID Val | $98.16 \pm 0.52$ | $98.97 \pm 0.31$ | $99.16 \pm 0.23$ | $99.60 \pm 0.29$ | $99.97 \pm 0.06$ | $99.97 \pm 0.04$ | $99.97 \pm 0.06$ |
| OOD Test | $23.71 \pm 0.11$ | $31.33 \pm 0.19$ | $35.47 \pm 0.20$ | $48.88 \pm 0.29$ | $73.97 \pm 0.20$ | $86.00 \pm 0.23$ | $85.82 \pm 0.37$ |

(b) Comparison across preconditioning exponents $p$.

## F.2 ROBUSTNESS OF THE $P^{(0)}$-ISOTROPIC INITIALIZATION ASSUMPTION

Figs. 8 and 9 present the training trajectories at each SNR level corresponding to Fig. 1, showing the results when the first layer is initialized with $P^{(0)}$ following the model setup in Theorems 1 and 2. The outcomes are consistent with those obtained under standard isotropic initialization in Sections 4.1 and 4.1.2, as well as Appendix F.1, suggesting that the initialization assumption can be empirically relaxed and that our claims hold under generic isotropic initialization.

## F.3 OOD GENERALIZATION

Table 2 presents the table corresponding to Figures 2a and 2b.

## F.4 TRAINING-FREE ESTIMATION OF TASK-RELEVANT SPECTRAL COMPONENTS

To demonstrate the practical implications of our framework, we explore a method for selecting the optimal optimizer—specifically in terms of generalization performance—prior to training. Drawing inspiration from Abbe et al. (2022), we propose estimating which spectral components are effective for label prediction using the inference of the model at initialization.

Abbe et al. (2022) defined the Initial Alignment (INAL) as the expected squared correlation between the target function $f^*$ and the output of the neural network at initialization $f_{\theta_0}$ (More precisely, they calculate the following INAL for each neuron and take the maximum value.:

$$\text{INAL}(f^*, f) = \mathbb{E}_{\theta_0}[\langle f^*, f_{\theta_0} \rangle^2]. \tag{21}$$

They established that a higher INAL serves as a strong indicator of learnability and subsequent generalization performance, signifying that the model at initialization already possesses components effective for label prediction.

Extending this concept to our spectral analysis, we calculate the INAL *separately* for each eigen-component of the input to identify specific important directions. We applied this method to the synthetic datasets (High and Low cases) described in the main text. The results are summarized in Tables 3 and 4.

In both cases, we observed a distinct spike in the INAL value relative to the median specifically for the components carrying the informative signal (Component 0 for Case High, and Component 9 for Case Low). These results suggest that task-relevant spectral components can be estimated in a training-free manner. We believe this validates our framework's potential to guide the selection of the optimal optimizer (i.e., the optimal preconditioner power $p$) by matching the spectral bias to the task structure, thereby replacing heuristic choices with a principled, task-aligned approach.

Table 3: INAL per component for **Case High** (where the unique high-variance component is effective). Component 0 exhibits a significantly higher alignment ratio compared to the median.

| Component | INAL (mean $\pm$ std) | Relative ratio (vs. median) |
|:---:|:---:|:---:|
| **0** | **0.81 $\pm$ 1.16** | **1.95$\times$** |
| 1 | 0.44 $\pm$ 0.60 | 1.05$\times$ |
| 2 | 0.42 $\pm$ 0.60 | 1.01$\times$ |
| 3 | 0.44 $\pm$ 0.60 | 1.05$\times$ |
| 4 | 0.42 $\pm$ 0.59 | 1.00$\times$ |
| 5 | 0.43 $\pm$ 0.61 | 1.02$\times$ |
| 6 | 0.40 $\pm$ 0.60 | 0.95$\times$ |
| 7 | 0.41 $\pm$ 0.58 | 1.00$\times$ |
| 8 | 0.40 $\pm$ 0.57 | 0.98$\times$ |
| 9 | 0.41 $\pm$ 0.58 | 1.00$\times$ |

Table 4: INAL per component for **Case Low** (where the unique low-variance component is effective). Component 9 exhibits a significantly higher alignment ratio compared to the median.

| Component | INAL (mean $\pm$ std) | Relative ratio (vs. median) |
|:---:|:---:|:---:|
| 0 | 0.018 $\pm$ 0.024 | 1.06$\times$ |
| 1 | 0.017 $\pm$ 0.023 | 1.01$\times$ |
| 2 | 0.017 $\pm$ 0.024 | 0.99$\times$ |
| 3 | 0.016 $\pm$ 0.022 | 0.96$\times$ |
| 4 | 0.017 $\pm$ 0.024 | 1.00$\times$ |
| 5 | 0.016 $\pm$ 0.023 | 0.98$\times$ |
| 6 | 0.018 $\pm$ 0.025 | 1.08$\times$ |
| 7 | 0.017 $\pm$ 0.024 | 1.03$\times$ |
| 8 | 0.016 $\pm$ 0.024 | 0.97$\times$ |
| **9** | **0.030 $\pm$ 0.042** | **1.81$\times$** |

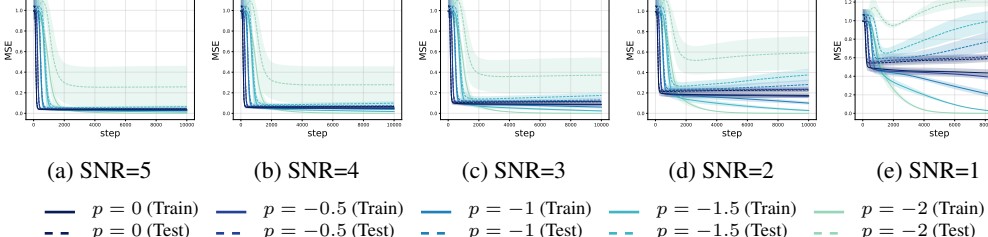

(a) SNR=5      (b) SNR=4      (c) SNR=3      (d) SNR=2      (e) SNR=1

—— $p = 0$ (Train)  —— $p = -0.5$ (Train)  —— $p = -1$ (Train)  —— $p = -1.5$ (Train)  —— $p = -2$ (Train)
-- $p = 0$ (Test)  -- $p = -0.5$ (Test)  -- $p = -1$ (Test)  -- $p = -1.5$ (Test)  -- $p = -2$ (Test)

Figure 8: Training and test performance trajectories for each SNR level with the exact covariance preconditioner for different $p$ on Case High. The first layer is initialized with an $P^{(0)}$-isotropic initialization.

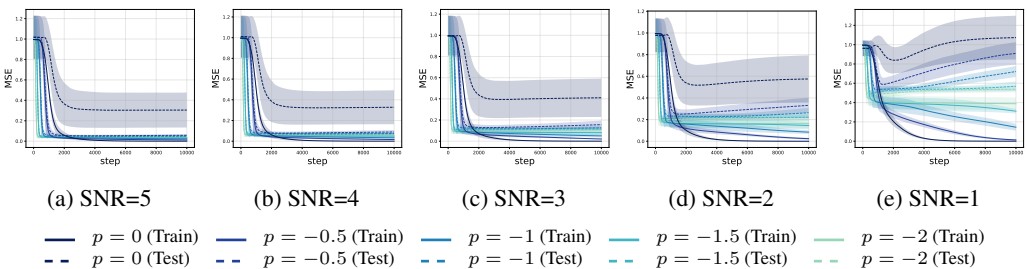

(a) SNR=5      (b) SNR=4      (c) SNR=3      (d) SNR=2      (e) SNR=1

—— $p = 0$ (Train)  —— $p = -0.5$ (Train)  —— $p = -1$ (Train)  —— $p = -1.5$ (Train)  —— $p = -2$ (Train)
-- $p = 0$ (Test)  -- $p = -0.5$ (Test)  -- $p = -1$ (Test)  -- $p = -1.5$ (Test)  -- $p = -2$ (Test)

Figure 9: Training and test performance trajectories for each SNR level with the exact covariance preconditioner for different $p$ on Case Low. The first layer is initialized with an $P^{(0)}$-isotropic initialization.

