# OpenReview forum: "How Does Preconditioning Guide Feature Learning in Deep Neural Networks?"
_ICLR.cc/2026/Conference — Submitted to ICLR 2026_

### Official Review · Reviewer_knWb · 2025-10-21

**Soundness:** 3
**Presentation:** 3
**Contribution:** 1
**Rating:** 2
**Confidence:** 3

**Summary:**

The authors consider the effect of an unsupervised linear transformation, a so-called preconditioning matrix, applied to the data on feature learning in the single-index model case. In particular, the authors consider the specific preconditioning matrix is a spectral transformation of the input covariance. The authors show theoretically that the first layer weights, with Gaussian initialization, erase information about the original basis representation of the data. The authors then perform experiments showing that aligning input covariance directions with the task-relevant direction improves performance, and, similarly, unaligning the data with task-direction worsens performance. They show this effect in the settings of noise and transfer learning. They provide observations of the effect of various observations on two synthetic settings with their preconditioning approach.

**Strengths:**

-	The paper addresses fundamental questions of deep learning: (1) how do neural networks learn features from data, and (2) what is the impact of optimization method on feature learning?
-	The experiments are thorough and are presented cleanly.

**Weaknesses:**

The observations in the paper are frankly not surprising at all and are relatively shallow in nature. First, their form of preconditioning is equivalent to viewing the effect of the covariance of the data on generalization, as their preconditioning matrix is neither trained nor supervised. This setting would be much more interesting in those cases or if preconditioning were studied in parameter space.
Then, it is not surprising that reducing the variance of the data in the directions that are relevant for prediction would worsen performance as you are emphasizing unimportant direction and placing the burden on the neural network to unlearn this preconditioning. This “unlearning” of unimportant directions has been studied much more precisely both theoretically and empirically in many of the prior works on feature learning (e.g. [1]).

[1] Damian, Lee, Soltanolkotabi. “Neural Networks can Learn Representations with Gradient Descent”, 2022.

The most interesting part of the paper to me is the experiments observing the comparison of various optimizers on their single index task across exponents for their preconditioning. However, the results for this setting are presented pretty much without explanation and matter-of-factly. It is not clear what the reader should take away from these experiments.

**Questions:**

None

---

> ### Author Response · Authors · 2025-11-21
> **Response to reviewer knWb**
>
> # Regarding weakness 1
>
> We sincerely thank the reviewer for their insightful comments and for pointing out the connection to prior work on feature learning.
>
> First, we would like to emphasize that while current research on optimizers heavily prioritizes accelerating convergence, the question of how different preconditioning matrices affect generalization performance—and the feature learning that supports it—is often overlooked. Our primary contribution is to provide a step toward clarifying this unexplored aspect.
>
> We acknowledge that in practice, full-parameter preconditioning matrices change dynamically during training and may exhibit complex behaviors. However, for the first-layer neuron-wise Hessian, which is the focus of our study, the matrix consistently takes the form of a weighted covariance matrix at any point in training (please see Appendix C). Therefore, we wish to emphasize that our formulation remains grounded in the actual optimization process and does not significantly deviate from reality.
>
> Regarding the "unlearning" of unimportant directions, while it would be ideal to identify unimportant directions with certainty beforehand, determining them prior to training is non-trivial, as they vary depending on the task. We believe our contribution lies in systematizing the generalization performance of optimizers—which has historically been treated implicitly—under such conditions where the optimal direction is not known *a priori*. Regarding reference [1], we have revised the manuscript to explicitly cite it in the main text.

---

> ### Author Response · Authors · 2025-11-21
> **Response to reviewer knWb**
>
> # Regarding weakness 2
>
> We appreciate your interest in the results of the comparison of various optimizers.
>
> Underlying this analysis is the premise that each optimizer corresponds to preconditioning with a specific power $p$. Broadly speaking, second-order methods (such as L-BFGS, K-FAC, AdaHessian, and Sophia-H) correspond to $p = -1$, Adam to $p = -1/2$, GD to $p = 0$, and SAM to $p = 1$. The experiments in Figure 2 serve to verify the validity of this mapping. For instance, when high-variance components are important (and low-variance ones are unimportant), theory suggests that a larger $p$ leads to better generalization performance. Our experiments with actual optimizers confirmed that this dependency on $p$ is preserved. Conversely, when low-variance components are important, the opposite trend was confirmed. In conclusion, the analysis derived from our formulation holds true for actual optimizers: if high-variance components are important, SAM or GD will generalize better, whereas if low-variance components are important, second-order methods will generalize better.
>
> Furthermore, to demonstrate the practical implications of our framework, we explore a method for selecting the optimal optimizer---specifically in terms of generalization performance---prior to training. Drawing inspiration from [2], we propose estimating which spectral components are effective for label prediction using the inference of the model at initialization.
>
> [2] defined the Initial Alignment (INAL) as the expected squared correlation between the target function $f^*$ and the output of the neural network at initialization $f_{\theta_0}$ (More precisely, they calculate the following INAL for each neuron and take the maximum value.:
>
> \begin{equation}
> \text{INAL}(f^\star, f_{\theta_0}) = \mathbb{E}[\langle f^\star, f_{\theta_0} \rangle^2].
> \end{equation}
>
> They established that a higher INAL serves as a strong indicator of learnability and subsequent generalization performance, signifying that the model at initialization already possesses components effective for label prediction.
>
> Extending this concept to our spectral analysis, we calculate the INAL \textit{separately} for each eigen-component of the input to identify specific important directions. We applied this method to the synthetic datasets (\setting{High} and \setting{Low} cases) described in the main text. The results are summarized in Tables below.
>
> In both cases, we observed a distinct spike in the INAL value relative to the median specifically for the components carrying the informative signal (Component 0 for Case High, and Component 9 for Case Low).
>
> *1. High (where the unique high-variance component is effective)*
>
> | Component | INAL  | Relative ratio (vs. median) |
> | :--- | :---: | :---: |
> | **0** | **0.8117 ± 1.1558** | **1.95×** |
> | 1 | 0.4377 ± 0.6016 | 1.05× |
> | 2 | 0.4208 ± 0.5996 | 1.01× |
> | 3 | 0.4383 ± 0.6039 | 1.05× |
> | 4 | 0.4156 ± 0.5919 | 1.00× |
> | 5 | 0.4259 ± 0.6095 | 1.02× |
> | 6 | 0.3967 ± 0.5974 | 0.95× |
> | 7 | 0.4140 ± 0.5842 | 1.00× |
> | 8 | 0.4060 ± 0.5724 | 0.98× |
> | 9 | 0.4149 ± 0.5833 | 0.998× |
>
> *2. Low (where the unique low-variance component is effective)*
>
> | Component | INAL (mean ± std) | Relative ratio (vs. median) |
> | --- | :---: | :---: |
> | 0 | 0.01783 ± 0.02449 | 1.06× |
> | 1 | 0.01686 ± 0.02330 | 1.01× |
> | 2 | 0.01652 ± 0.02358 | 0.99× |
> | 3 | 0.01608 ± 0.02187 | 0.96× |
> | 4 | 0.01668 ± 0.02442 | 1.00× |
> | 5 | 0.01641 ± 0.02342 | 0.98× |
> | 6 | 0.01810 ± 0.02464 | 1.08× |
> | 7 | 0.01728 ± 0.02398 | 1.03× |
> | 8 | 0.01621 ± 0.02407 | 0.97× |
> | **9** | **0.03035 ± 0.04174** | **1.81×** |
>
>
> These results suggest that task-relevant spectral components can be estimated in a training-free manner. We believe this validates our framework's potential to guide the selection of the optimal optimizer (i.e., the optimal preconditioner power $p$) by matching the spectral bias to the task structure, thereby replacing heuristic choices with a principled, task-aligned approach.
>
> We added this results and discussion to the appendix in our manuscript.
>
> [2] Abbe, Emmanuel, et al. "An initial alignment between neural network and target is needed for gradient descent to learn." International Conference on Machine Learning. PMLR, 2022.

---

### Official Review · Reviewer_oPxn · 2025-10-31

**Soundness:** 4
**Presentation:** 3
**Contribution:** 3
**Rating:** 6
**Confidence:** 1

**Summary:**

This paper investigates how preconditioning affects generalization, arguing that its primary role is to control and bias the feature learning process.
The central argument is that the preconditioner (e.g., from Adam or K-FAC) acts as a filter, inducing a "spectral bias" by forcing the model to only see the data through a specific, preconditioned Gram matrix.
The authors model this by instantiating the preconditioner as the $p$-th power of the input covariance, $\Sigma_X^p$, where the exponent $p$ controls this bias: large $p$ emphasizes high-variance features, while small $p$ emphasizes low-variance features.
Using synthetic "teacher-student" experiments, the paper demonstrates that generalization is only achieved when the optimizer's spectral bias is aligned with the teacher's signal.

**Strengths:**

This paper investigates an interesting question with a reasonable approach. The results appear to me sound and convincing.

**Weaknesses:**

W1. The experiments are done with synthetic data. Is there any way to test this idea on more realistic setups?

Other than this, there are no obvious weaknesses to me at the moment, though I unfortunately cannot strongly endorse the paper as I am a non-expert reviewer (see confidence).

**Questions:**

Q1. What are the implications of your findings for how one might set the hyperparameters for off-the-shelf optimizers like Adam?

---

> ### Author Response · Authors · 2025-11-21
> **Response to reviewer oPxn**
>
> # Regarding weakness 1
>
> We thank the reviewer for the constructive suggestion.
>
> First, we would like to emphasize that our validation is not limited to purely synthetic data; as detailed in Section 4.2, we conducted experiments using real image data (MNIST). Crucially, in this setting, we compared realistic optimizers used in practice rather than just theoretical proxies, and we obtained strong results that are consistent with our theoretical framework.
>
> Furthermore, rigorously verifying our theory requires a priori knowledge of which eigen-components of the input are effective for label prediction. In more realistic datasets like ImageNet, this underlying alignment between the data spectrum and the target function is unknown, making it difficult to strictly guarantee consistency between experimental results and the theory. For this reason, we considered such large-scale benchmarks to be outside the scope of this work, prioritizing a controlled setting where the causal link between spectral bias and generalization could be clearly established.

---

> > ### Author Response · Authors · 2025-11-21
> > **Response to reviewer oPxn**
> >
> > # Regarding question 1
> >
> > We appreciate your inquiry regarding the hyperparameters of the optimizer.
> >
> > Our findings offer two concrete implications for practitioners using off-the-shelf optimizers like Adam.
> > 1. $\epsilon$ as a Spectral Knob: While standard Adam applies a fixed preconditioning power roughly corresponding to $p=-0.5$ (inverse square root scaling), the stability parameter $\epsilon$ allows for interpolation between this behavior and standard Gradient Descent ($p=0$). Specifically, as $\epsilon$ increases, the preconditioning effect diminishes. Our theory suggests that for tasks relying on high-variance features (dominant signals), setting a larger $\epsilon$ may improve generalization by shifting the behavior closer to GD. Conversely, for tasks depending on low-variance features (subtle details), a small $\epsilon$ is essential to maintain the whitening effect.
> > 2. Optimizer Selection as Spectral Tuning: Our work suggests that the choice of optimizer algorithm should be viewed as selecting the spectral exponent $p$. If Adam ($p \approx -0.5$) fails to generalize well, practitioners should consider switching the optimizer class based on the suspected task structure: moving to SGD ($p=0$) if the model needs to focus on dominant high-variance features, or to more aggressive second-order methods ($p \approx -1$) if the model fails to capture subtle, low-variance features.

---

> > > ### Comment · Reviewer_oPxn · 2025-11-27
> > >
> > > Thank you for the clarification on real-world data and for your detailed answer on Adam. I wish to maintain my score due to the Reviewer Confidence reasons listed above.

---

### Official Review · Reviewer_vLdJ · 2025-11-01

**Soundness:** 3
**Presentation:** 3
**Contribution:** 2
**Rating:** 6
**Confidence:** 4

**Summary:**

In this paper, the role of preconditioning (specifically by powers of the input covariance matrix) on feature learning is studied. In particular, varying the power of the preconditioner affects the quantitative properties of the learned features via imposing a spectral bias of varying strength. The implications are demonstrated via noise robustness properties, generalization out-of-distribution, and transfer learning.

**Strengths:**

Overall, I think this is a solid paper that presents an interesting perspective on a scale of preconditioning that (as far as I know) has not been studied in prior work. The key theoretical result demonstrates the model evolution given a training batch can be fully captured by some salient statistics. These statistics clearly delineate how the "spectral bias" arises in feature learning. The experiments comparing multiple optimizers, notably SAM (corresponding approximately to the preconditioner power $p=1$), GD ($p=0$) and various second order optimizers ($p=-1$) serves as an interesting empirical reflection of the theoretical predictions.

**Weaknesses:**

In my opinion, the main weaknesses of the paper is the extrapolation from the provided theory and empirical claims. Notably, the theory in the paper focuses on one source of preconditioning via the input covariance, the experiments consider a wide range of preconditioned optimizers lumped under "second-order methods". Regardless of the empirical results, the qualitative differences between the curvature matrices approximated across the different methods is large enough that I'd be hesitant to attribute observed trends to the proposed spectral biases, even if for certain cases there is certainly alignment between different curvature matrices. Some more concrete derivations (even in simple one/two-layer linear networks) to show the relation between the curvature matrices between the different optimizers would help clarify this point.

It should also be noted that some optimizers in the "second-order" $p=-1$ category (e.g. AdaHessian and Adam) take a square-root power of the preconditioner, which suggests $p=-1/2$ would be more accurate there. Also, is it supported why SAM corresponds to $p=1$?

Lastly, though the takeaway in Section 4.3 that "second-order" optimization has a role in transfer learning, this observation is not necessarily novel, see e.g. [1], where the proposed intervention is indeed to pre-condition by the inverse input covariance matrix to enable feature/transfer learning. Furthermore, considering the specific discussion about KFAC is rather light, it might be good to note the corresponding "right-side" preconditioner KFAC induces on the input layer weight matrix is also the inverse input covariance.

[1] Zhang et al. "Sample-Efficient Linear Representation Learning from Non-IID Non-Isotropic Data"

**Questions:**

Please see Weaknesses above.

---

> ### Author Response · Authors · 2025-11-21
> **Response to reviewer vLdJ**
>
> # Response to reviewer vLdJ
>
> We appreciate your proposal for a more in-depth analysis of second-order methods.
>
> Here we summarize how close each second-order method is to preconditioning at the strict $p=-1$ for each of them.
>
> **K-FAC**. K-FAC approximates the FIM layer-wise by decomposing it into the Kronecker product of output-gradient and input covariances:
>
> $$
> \boldsymbol{F} \approx \mathbb{E}\left[ \nabla_{\boldsymbol{z}} \mathcal{L} (\nabla_{\boldsymbol{z}} \mathcal{L})^\top \right] \otimes \mathbb{E}\left[ \boldsymbol{x} \boldsymbol{x}^\top \right] := \boldsymbol{G} \otimes \boldsymbol{A}.
> $$
>
> Crucially, the input factor $\boldsymbol{A} = \mathbb{E}[\boldsymbol{x} \boldsymbol{x}^\top]$ coincides exactly with the input covariance matrix $\Sigma_X$. Consequently, the preconditioned update rule applies the inverse of this covariance to the input side of the weight gradient:
>
> $$
> \Delta \boldsymbol{W} = \boldsymbol{A}^{-1} (\nabla_{\boldsymbol{W}} \mathcal{L}) \boldsymbol{G}^{-1} = \boldsymbol{\Sigma_X}^{-1} (\nabla_{\boldsymbol{W}} \mathcal{L}) \boldsymbol{G}^{-1}.
> $$
>
> Since the input-side preconditioner directly determines the similarity metric affecting inference in our framework, this left-multiplication by $\boldsymbol{\Sigma_X}^{-1}$ formally aligns K-FAC with the whitening regime where $p=-1$. Unlike diagonal approximations, K-FAC retains the dense structure of $\boldsymbol{\Sigma_X}$, making it the implementation closest to the theoretical ideal of $p=-1$ among the evaluated methods.
>
> **AdaHessian and Sophia-H.** These methods rely on diagonal approximations of the Hessian applied with an inverse power. While they aim for a Newton-like update, they strictly correspond to $p=-1$ only if the input features are independent (i.e., the covariance matrix is diagonal). In practice, neglecting off-diagonal components distances them from the exact spectral bias of $p=-1$ compared to K-FAC. Furthermore, they differ in their handling of small eigenvalues: AdaHessian employs soft damping, resulting in a smooth transition between $p=-1$ and $p=0$ as eigenvalues decrease. In contrast, Sophia-H utilizes a hard clipping mechanism, creating a distinct boundary where directions with curvature below a threshold effectively revert to $p=0$.
>
> **L-BFGS.** L-BFGS approximates the inverse Hessian using a history of the past $m$ gradient updates. It captures curvature information ($p \approx -1$) specifically along the directions spanned by these recent updates. However, for the vast majority of the parameter space orthogonal to the history, it retains unit curvature ($p=0$). Consequently, L-BFGS operates as a hybrid method, mixing the behaviors of $p=-1$ and $p=0$.
>
> **Adam.** Adam scales gradients by the inverse square root of the second moment estimate, which roughly corresponds to $p=-1/2$. This formulation places its spectral bias closer to GD than to the pure second-order methods discussed above.
>
> **Summary.** Based on this analysis, we can order the optimizers by their proximity to the theoretical $p=-1$ regime: K-FAC is the nearest, followed by the diagonal/hybrid approximations (AdaHessian, Sophia-H, L-BFGS), with Adam being the furthest and closest to GD. The experimental results in Figure 2a are consistent with this hierarchy, demonstrating that optimizers with a stronger structural fidelity to $p=-1$ exhibit the most distinct spectral biases (e.g., excelling when low-variance features are invariant, as seen in the "Invariant Noise" setting).
>
> We have added the above discussion to the appendix and highlighted it in red.

---

> ### Author Response · Authors · 2025-11-21
> **Response to reviewer vLdJ**
>
> # Regarding weakness 2
>
> Thank you for bringing up the concern about the value of $p$.
>
> **With respect to Adahessian**, since it takes the square root of the “square” of the Hessian in its implementation, we hold the belief that the correspondence $p = -1$ is more suitable.
>
> **Concerning Adam**, you are indeed correct, and $p = -1/2$ would have been a more appropriate choice. We have taken this matter into consideration and highlighted it by revising the paper.
>
> **Regarding SAM**, it can be associated with p=1 through the following simple derivation.
>
> Recall the gradient computation in SAM, denoted as $\boldsymbol{g}_{\text{SAM}}$, which involves computing the gradient at a perturbed parameter location to maximize the loss locally:
>
> \begin{equation}
> \boldsymbol{g}_{\text{SAM}} = \nabla\mathcal{L}\left(\boldsymbol{\theta} + \frac{\rho}{||\nabla\mathcal{L}(\boldsymbol{\theta})||}\nabla\mathcal{L}(\boldsymbol{\theta})\right),
> \end{equation}
>
> where $\rho$ is the neighborhood size. Let us define the perturbation vector $\boldsymbol{\delta}$ as:
>
> \begin{equation}
> \boldsymbol{\delta} = \frac{\rho}{||\nabla\mathcal{L}(\boldsymbol{\theta})||}\nabla\mathcal{L}(\boldsymbol{\theta}).
> \end{equation}
>
> Assuming that $\rho$ is sufficiently small, we can apply a first-order Taylor expansion of the gradient $\nabla\mathcal{L}(\boldsymbol{\theta} + \boldsymbol{\delta})$ around $\boldsymbol{\theta}$. Let $\boldsymbol{H} = \nabla^2\mathcal{L}(\boldsymbol{\theta})$ denote the Hessian matrix of the loss function at $\boldsymbol{\theta}$. The approximation proceeds as follows:
>
> \begin{align}
> \boldsymbol{g}_{\text{SAM}} &\approx \nabla\mathcal{L}(\boldsymbol{\theta}) + \boldsymbol{H} \boldsymbol{\delta} \nonumber = \nabla\mathcal{L}(\boldsymbol{\theta}) + \boldsymbol{H} \left( \frac{\rho}{||\nabla\mathcal{L}(\boldsymbol{\theta})||}\nabla\mathcal{L}(\boldsymbol{\theta}) \right) \nonumber = \nabla\mathcal{L}(\boldsymbol{\theta}) + \frac{\rho}{||\nabla\mathcal{L}(\boldsymbol{\theta})||} \boldsymbol{H} \nabla\mathcal{L}(\boldsymbol{\theta}).
> \end{align}
>
> By factoring out the gradient term $\nabla\mathcal{L}(\boldsymbol{\theta})$, we obtain the final approximate form:
>
> \begin{equation}
> \boldsymbol{g}_{\text{SAM}} \approx \left( I + \frac{\rho}{||\nabla\mathcal{L}(\boldsymbol{\theta})||} \boldsymbol{H} \right) \nabla\mathcal{L}(\boldsymbol{\theta}).
> \end{equation}
>
> This derivation illustrates that the effective gradient used in SAM is approximately equivalent to the standard gradient preconditioned by the matrix $\left( I + \frac{\rho}{||\nabla\mathcal{L}(\boldsymbol{\theta})||} \boldsymbol{H} \right)$. Therefore, SAM corresponds to $p = 1$ in our formulation.
>
> We have also included this derivation in the appendix.

---

> ### Author Response · Authors · 2025-11-21
> **Response to reviewer vLdJ**
>
> # Regarding weakness 3
>
> We thank the reviewer for the valuable feedback and for pointing out the relevant work in [1].
>
> **Regarding Transfer Learning and Prior Work**: We explicitly acknowledge that our findings in Section 4.3 align with the conclusions of [1], which advocates for preconditioning by the inverse input covariance matrix. In our revised manuscript, we have added a citation to [1] to properly credit this alignment. However, we would like to emphasize that our work derives this conclusion as part of a systematic analysis of preconditioning. Rather than focusing solely on the benefits of whitening, we discuss the relationship between preconditioning and feature learning more broadly by sweeping across the spectral bias $p$. We position the effectiveness of inverse covariance preconditioning ($p=-1$) within this wider spectrum, highlighting the trade-offs between source-task specialization and target-task transferability.
>
> **Regarding K-FAC**: We appreciate the reviewer's note on the structure of K-FAC. We have revised the discussion to be more precise. While K-FAC indeed induces inverse covariance preconditioning on the weights, we have clarified that our theoretical focus is specifically on the input-side matrix.

---

### Official Review · Reviewer_xAve · 2025-11-23

**Soundness:** 3
**Presentation:** 3
**Contribution:** 2
**Rating:** 4
**Confidence:** 3

**Summary:**

This paper extends information-theoretic results on data whitening to a general preconditioning framework, investigating how preconditioned gradient descent affects feature learning and generalization in neural networks. The main theoretical contribution establishes that all input information available to the model flows exclusively through the preconditioned Gram matrix
$ \(\mathbf{G}_P = \mathbf{X}^T \mathbf{P} \mathbf{X}\) $, where $\(\mathbf{P}\)$ is a positive semi-definite preconditioner. When instantiated as $\(\mathbf{P} = \boldsymbol{\Sigma}_X^p\)$ (the $\(p\)$-th power of input covariance), the Gram matrix's singular values scale as $ \(s_r^{2(p+1)}\) $, creating what the authors term $ \textit{spectral bias} $: higher exponents $\(p\)$ amplify high-variance input directions, while lower values (including negative) emphasize low-variance directions. It is formalization of quite well-known fact: the preconditioner induces a metric on input space; features aligned with that metric are preferentially learned.

The paper leaves the reader with clear (theoretical) message: for good generalization, one should match the preconditioner's spectral bias to the task structure. The paper provides somewhat concrete evidence through three experimental settings: robustness to label noise, out-of-distribution generalization on corrupted MNIST, and transfer learning across synthetic tasks.

Mathematically, Theorems 1 and 2 establish conditional independence relations showing that hidden layer activations and deep parameters remain conditionally independent of raw inputs given the Gram matrix history and labels. The paper naturally extends existing research work: Wadia et al.'s (2021) analysis of whitening to arbitrary preconditioners.

While the paper is theoretically sound, I have some concerns regarding the clarity of presentation. The main issues is empirical work - which is more in-detailed described in weaknesses section.

**Strengths:**

S1: Clear Research Objective and Practical Relevance

The paper addresses a well-motivated question: how does preconditioning affect generalization, beyond its known role in accelerating convergence? The research objective is clear how to understand interplay between preconditioning and feature learning in deep learning. This is genuinely important because optimizer choice is typically made heuristically. By characterizing preconditioners through spectral bias, the paper provides a principled lens for understanding why different algorithms (SAM, K-FAC, L-BFGS, AdaHessian) exhibit different generalization properties on the same task. This contribution may have immediate relevance for practitioners.

S2: Reconciliation of Conflicting Prior Work

The paper elegantly resolves tension in the literature: Wadia et al. (2021) argued whitening is harmful to generalization by destroying information in the Gram matrix. Yet recent work shows benefits in specific contexts. This paper explains both via the alignment principle: generalization depends on whether the preconditioner's spectral bias matches where the true task-relevant signal lives spectrally. By extending beyond whitening to a continuous family parametrized by exponent $\(p\)$, the paper moves from binary "positive/negative'' judgment to task-dependent analysis.

S3 Elegant Mathematical Framework

Theorems 1 and 2 employing conditional independence via data-processing inequality are well-executed and provide a unifying perspective. The spectral characterization (Eq 7) elegantly shows how the exponent $\(p\)$ directly modulates singular value scaling: $\(s_r^{2(p+1)}\)$. This transparency is valuable for comprehensive understanding.

S4: Thoughtfully Designed Experiments with Partially Compelling Evidence

The experimental design is illustrative and well-conceived:

a) Section 4.1 cleanly isolates spectral bias effects in synthetic settings with explicit signal placement (Cases High vs Low).
b) Section 4.2 provides the empirical result (IMO standing out): reversible optimizer rankings across two OOD scenarios (Figure 2). The ranking flip between invariant-digit and invariant-noise cases leads direct evidence that different optimizers have opposite spectral biases.
c) Section 4.3 offers a (limited, but still a bit novel taking into consideration the perspective) insight (negative $\(p\)$ optimal for transfer) with practical implications.

S5: Strong Presentation and Organization

The paper is well-structured with clear claims, logical flow, and effective visualizations.

**Weaknesses:**

[MAJOR]

W1: Severe Scalability Limitations and Dataset Concerns

The experimental validation is limited in scope:

i. Synthetic experiments (4.1): Use only 200 training samples with dimension 10, trained on a two-layer MLP with hidden dimension 256. While synthetic control is valuable for proof-of-concept, this extreme case relative to sample size is far removed from realistic settings.

ii. Moreover, Figure 1 (top row) shows training curves that do not appear fully converged across most hyperparameter configurations. Since the paper compares generalization across conditions with different convergence levels, interpreting differences as evidence for the alignment principle becomes problematic. Fair comparison would require either training to convergence or comparing generalization at equal training loss/accuracy levels. The authors should clarify convergence criteria (definitely not constant optimization number of steps as this paper lies in optimization landscape) and provide terminal training&validation loss values given the convergence.

iii. MNIST experiments (4.2): Use approximately 1k training samples from MNIST (not stated clearly, I inferred it). While the controlled noise injection enables hypothesis testing, the result is barely above toy-problem scale. The split into spurious and invariant components is artificial and may not reflect realistic distribution shifts encountered in practice - there are other, more natural variants of noise injection in computer vision community.

iv. Transfer learning (4.3) Validates on a single symmetric synthetic task. For continual learning claims, two sequential tasks are definitely insufficient; typically we examine 5--20 tasks minimum. Real multi-task transfer learning (e.g., ImageNet pretraining to downstream tasks) would substantially strengthen claims.

W3: Substantial Theory-Practice Gap

Some assumptions required by the theory are violated in experiments:

i. $P(0)$-isotropic initialization: Theorem 1 requires weights to be initialized as$ \(\mathbf{w}_j^{(0)} \sim \mathcal{N}(\mathbf{0}, \sigma^2 \mathbf{P}^{(0)})\) $, meaning the initial covariance matrix must match the preconditioner structure. This requires:
- Pre-computing the full covariance matrix $ \(\mathbf{P}^{(0)} = \boldsymbol{\Sigma}_X^p\) $ before training (at least SVD is needed, right?)
- Custom initialization departing from standard schemes (for instance Xavier).

According to my best knowledge, these experiments do not appear to use this initialization; they likely employ standard initialization. Does it really matter in the end? If so, please empirically validate.

ii. Stochasticity: I am uncertain, but I guess that the theory assumes deterministic training. Experiments use mini-batch SGD, which may disrupt the deterministic Gram matrix bottleneck through finite-sample fluctuations. Same question as above, does it matter?

W4: Limited/Unrealistic Scope of The Proposed Setting

i. Single-index teacher limitation: The quantitative claims (Sec 3.3) assume labels follow $ \(y = h^*(\sum_r \alpha_r \beta_r) + \epsilon\) $, a one-dimensional structure in label space. Real tasks exhibit multi-dimensional structure. The paper provides no analysis of whether insights extend to multi-index teachers, severely limiting generalizability of Theorems 1 and 2 beyond this restrictive setting.

ii. [MINOR] First-layer-only preconditioning: Analysis applies only to preconditioned first-layer updates. Modern networks use layer-wise preconditioning (K-FAC, Sophia) as in one of the cited work Zhang et al. 2025. Probably this spectral bias does not propagate through depth.

[Minor Weaknesses]

W5: Experimental Design Ambiguities

i. (Sec 4.2.1) The exact experimental procedure for OOD evaluation is not entirely clear from the main text. It appears that: (a) the model is trained on data with digit+class-specific noise (one of them, right?), (b) at test time, either the noise or digit is changed to reverse which component is "spurious". Additionally, Figure 2 error bars show standard deviation only for in-distribution evaluation; reporting error bars for OOD accuracy would be more informative.

ii. Incomplete Analysis of Related Work

The paper cites but does not deeply discus Zhang et al.'s findings on layer-wise preconditioning and feature learning (I guess here more work should be pointed out), which partially overlap with the insights in Sec 4.3.

iii. Code Availability Issues

While the authors provide code (I appreciate it however it is not required), significant usability issues limit reproducibility:
- 2 of 5 scripts run but produce only single train/test loss values (insufficient information/analysis without any context as it is on particular engineered data);
- 2 scripts yield runtime errors (TypeError in AdaHessian.step() );
- 1 script has data iterator issues even after straightforward fixes applied.

Hence I do not see the purpose of it. If code is provided, it should be functional. Otherwise, omitting the repository might be preferable.

W6: Although I find the structure, research questions/hypotheses outlined useful and clear, the mathematical part is quite convoluted and sometimes hard to follow.

**Questions:**

Q1: Can you provide a rigorous derivation connecting SAM's preconditioner to \(p = 1\), or should this be framed as an empirical observation validated by Figure 2b?

I am not convinced that SAM can be viewed as an approach with conditional power being 1 and here is why:
SAM: $ \mathbf{g}_{\text{SAM}} = \nabla L(\mathbf{\theta} + \boldsymbol{\delta}) $

where perturbation is given $ \boldsymbol{\delta} = \frac{\rho}{\|\nabla L(\mathbf{\theta})\|} \nabla L(\mathbf{\theta})\ $.
We may apply Taylor's approx.: $ \mathbf{g}_{\text{SAM}} \approx \nabla L(\mathbf{\theta}) + \mathbf{H} \boldsymbol{\delta} = \nabla L(\mathbf{\theta}) + \frac{\rho}{\|\nabla L\|} \mathbf{H} \nabla L(\mathbf{\theta}) $.
Hence for sufficiently small $\rho$ we obtain:
$ g \approx (\mathbf{I} - \rho \mathbf{H}) \nabla L(\mathbf{\theta}) $
It feels that: $ (\mathbf{I} - \rho \boldsymbol{\Sigma}_X)^{-1} \approx \boldsymbol{\Sigma}_X^{-1} $.
However, intuitively, I understand your claim that SAM emphasizes high-variance directions in the input space.

Q2: I do have question about averaging results in context of aggregating across initializations / random seeds. (related to W5, but not limitted)

Q3: For transfer learning: can you validate the $\(p = -1\)$ optimality on realistic multi-task settings (e.g., ImageNet pretraining to downstream tasks) rather than only synthetic scenarios?

Q4: How in practice, one can try to align spectral bias without requirement to having access to the teacher's signal.

---

> ### Author Response · Authors · 2025-11-29
> **Response to Reviewer xAve**
>
> ## Regarding The Scalability of Experiments (W1 i and iii)
>
> We appreciate the reviewer's feedback regarding the data scale in Section 4.1 and 4.2.
>
> While we acknowledge the simplified setting, we respectfully submit that this controlled environment is essential to strictly isolate and verify the theoretical mechanism proposed$-$specifically, how the alignment between the teacher signal and the input spectrum dictates generalization. In complex real-world datasets like CIFAR or ImageNet, **the spectral nature of the true teacher signal remains unknown, making it impossible to rigorously confirm whether the spectral alignment is actually occurring as predicted.** Consequently, while we establish the theoretical foundation here, we consider the development of methodologies to precisely identify and analyze these task-relevant spectral components within large-scale real-world datasets as a vital direction for future work.
>
> By contrast, our synthetic setup provides the precise causal control necessary to prove this relationship. Furthermore, Section 4.2 serves to bridge this theoretical proof to realistic settings; using MNIST allows us to move beyond purely synthetic data to a regime closer to real-world complexity while retaining identifiability of signal location, where we observe that practical optimizers behave entirely consistently with our theoretical framework.
>
> ## Regarding The Setting of Transfer Learning (W1 iv)
>
> We appreciate the reviewer’s perspective regarding continual learning benchmarks.
>
> While we fully agree that evaluating a long sequence of tasks is the standard protocol in continual learning research to assess cumulative knowledge retention, our study specifically addresses the "Pretraining-to-Finetuning" paradigm, which remains the dominant framework in practical transfer learning (e.g., foundation models adapting to downstream tasks). Our experimental design (Source Task $\to$ Target Task) was deliberately chosen to model this standard workflow. By utilizing this minimal canonical setting, we aimed to rigorously isolate and demonstrate the mechanism of how the spectral bias induced during pretraining facilitates or hinders adaptation to a subsequent downstream task, rather than evaluating stability over a long sequence of tasks.

---

> ### Author Response · Authors · 2025-11-29
> **Response to Reviewer xAve**
>
> ## Regarding The Convergence Level in Figure 1  (W1 ii)
>
> Thank you for pointing out concerns about the convergence level in Figure 1.
>
> First, we respectfully posit that the original fixed-step comparison is methodologically valid and that the models have effectively converged relative to the true signal. Specifically, we confirmed that all models presented in Figure 1 successfully reduced their training loss to the level of the expected noise variance ($E[\epsilon^2]$) within the allocated steps. In this context of noisy label learning, driving the training loss further toward zero signifies overfitting to the noise rather than learning the signal. As evidenced by the trajectories in the bottom row of Figure 1, models with poor spectral alignment are significantly more susceptible to fitting these noise components once the signal is learned, leading to severe overfitting compared to well-aligned models. Therefore, the reported performance differences at fixed steps do not reflect a failure to learn the signal, but rather clearly capture the varying degrees of resistance to noise overfitting governed by $p$, which strongly reinforces our central hypothesis.
>
> Nevertheless, to directly address the reviewer’s concern regarding fair comparison at strictly identical convergence levels, we have conducted an additional evaluation. Instead of a fixed step count, we recorded the test loss at the specific step where the training loss first reached the theoretical expected noise variance ($E[\epsilon^2]$). The results are presented in the table below:
>
> ### Case High
>
> | (SNR = 1) | p=-2.0 | p=-1.5 | p=-1.0 | p=-0.5 | p=0.0 |
> | :--- | :---: | :---: | :---: | :---: | :---: |
> | **Train** | 0.4979 ± 0.0743 | 0.4929 ± 0.0634 | 0.4902 ± 0.0630 | 0.4941 ± 0.0586 | 0.4917 ± 0.0525 |
> | **Test** | 0.8529 ± 0.1021 | 0.5956 ± 0.0621 | 0.5496 ± 0.0558 | 0.5167 ± 0.0437 | 0.5029 ± 0.0357 |
>
> | (SNR = 2) | p=-2.0 | p=-1.5 | p=-1.0 | p=-0.5 | p=0.0 |
> | :--- | :---: | :---: | :---: | :---: | :---: |
> | **Train** | 0.1984 ± 0.0430 | 0.1962 ± 0.0335 | 0.1960 ± 0.0314 | 0.1963 ± 0.0302 | 0.1981 ± 0.0296 |
> | **Test** | 0.4128 ± 0.1152 | 0.2406 ± 0.0372 | 0.2234 ± 0.0313 | 0.2084 ± 0.0233 | 0.2029 ± 0.0201 |
>
> | (SNR = 3) | p=-2.0 | p=-1.5 | p=-1.0 | p=-0.5 | p=0.0 |
> | :--- | :---: | :---: | :---: | :---: | :---: |
> | **Train** | 0.0950 ± 0.0240 | 0.0979 ± 0.0187 | 0.0985 ± 0.0173 | 0.0976 ± 0.0161 | 0.0990 ± 0.0161 |
> | **Test** | 0.2285 ± 0.0799 | 0.1212 ± 0.0216 | 0.1131 ± 0.0180 | 0.1050 ± 0.0131 | 0.1017 ± 0.0115 |
>
> | (SNR = 4) | p=-2.0 | p=-1.5 | p=-1.0 | p=-0.5 | p=0.0 |
> | :--- | :---: | :---: | :---: | :---: | :---: |
> | **Train** | 0.0587 ± 0.0170 | 0.0576 ± 0.0118 | 0.0584 ± 0.0107 | 0.0584 ± 0.0102 | 0.0583 ± 0.0099 |
> | **Test** | 0.1501 ± 0.0617 | 0.0718 ± 0.0131 | 0.0676 ± 0.0118 | 0.0632 ± 0.0090 | 0.0601 ± 0.0073 |
>
> | (SNR = 5) | p=-2.0 | p=-1.5 | p=-1.0 | p=-0.5 | p=0.0 |
> | :--- | :---: | :---: | :---: | :---: | :---: |
> | **Train** | 0.0374 ± 0.0139 | 0.0384 ± 0.0081 | 0.0384 ± 0.0073 | 0.0382 ± 0.0067 | 0.0385 ± 0.0067 |
> | **Test** | 0.1082 ± 0.0544 | 0.0478 ± 0.0091 | 0.0447 ± 0.0082 | 0.0417 ± 0.0062 | 0.0397 ± 0.0051 |
>
> ### Case Low
>
> | (SNR = 1) | p=-2.0 | p=-1.5 | p=-1.0 | p=-0.5 | p=0.0 |
> | :--- | :---: | :---: | :---: | :---: | :---: |
> | **Train** | 0.4903 ± 0.0533 | 0.4924 ± 0.0630 | 0.4936 ± 0.0770 | 0.4773 ± 0.0960 | 0.4982 ± 0.0743 |
> | **Test** | 0.5039 ± 0.0258 | 0.5185 ± 0.0309 | 0.5620 ± 0.0333 | 0.6469 ± 0.0680 | 1.0053 ± 0.0979 |
>
> | (SNR = 2) | p=-2.0 | p=-1.5 | p=-1.0 | p=-0.5 | p=0.0 |
> | :--- | :---: | :---: | :---: | :---: | :---: |
> | **Train** | 0.1998 ± 0.0287 | 0.1984 ± 0.0292 | 0.1985 ± 0.0308 | 0.1960 ± 0.0359 | 0.1881 ± 0.0436 |
> | **Test** | 0.2082 ± 0.0168 | 0.2143 ± 0.0175 | 0.2354 ± 0.0235 | 0.2783 ± 0.0414 | 0.6380 ± 0.1568 |
>
> | (SNR = 3) | p=-2.0 | p=-1.5 | p=-1.0 | p=-0.5 | p=0.0 |
> | :--- | :---: | :---: | :---: | :---: | :---: |
> | **Train** | 0.1000 ± 0.0159 | 0.0996 ± 0.0158 | 0.0999 ± 0.0159 | 0.0969 ± 0.0171 | 0.0961 ± 0.0271 |
> | **Test** | 0.1093 ± 0.0103 | 0.1126 ± 0.0113 | 0.1220 ± 0.0162 | 0.1468 ± 0.0248 | 0.4562 ± 0.1451 |
>
> | (SNR = 4) | p=-2.0 | p=-1.5 | p=-1.0 | p=-0.5 | p=0.0 |
> | :--- | :---: | :---: | :---: | :---: | :---: |
> | **Train** | 0.0588 ± 0.0096 | 0.0588 ± 0.0097 | 0.0585 ± 0.0092 | 0.0573 ± 0.0098 | 0.0574 ± 0.0189 |
> | **Test** | 0.0688 ± 0.0067 | 0.0705 ± 0.0076 | 0.0733 ± 0.0100 | 0.0907 ± 0.0176 | 0.3637 ± 0.1319 |
>
> | (SNR = 5) | p=-2.0 | p=-1.5 | p=-1.0 | p=-0.5 | p=0.0 |
> | :--- | :---: | :---: | :---: | :---: | :---: |
> | **Train** | 0.0385 ± 0.0062 | 0.0384 ± 0.0064 | 0.0382 ± 0.0059 | 0.0376 ± 0.0064 | 0.0371 ± 0.0145 |
> | **Test** | 0.0472 ± 0.0047 | 0.0485 ± 0.0055 | 0.0494 ± 0.0067 | 0.0621 ± 0.0134 | 0.3136 ± 0.1307 |
>
> **Crucially, these results demonstrate that even when controlling for the degree of convergence based on training loss, the performance ranking of different $p$ values remains unchanged, thereby confirming the robustness of our findings.**

---

> ### Author Response · Authors · 2025-11-29
> **Response to Reviewer xAve**
>
> ## Regarding The Gap of The Assumption for $P(0)$-isotropic initialization (W3 i)
>
> We appreciate your comment regarding the gap between the theory and experiments on initialization.
>
> We employed standard initialization in our main experiments to demonstrate that our theoretical insights hold under standard deep learning conditions, not because $P^{(0)}$-initialization is problematic. However, following your suggestion, we conducted an ablation study using the strict $P^{(0)}$-isotropic initialization. The results confirm that the initialization scheme has a negligible impact on the final outcome: the generalization performance ranking across different $p$ values remains unchanged. This indicates that our theoretical analysis is robust to initialization strategies.
>
> The table below reports the test loss at the step where the training loss first reaches the expected noise variance. As with standard initialization, the performance ranking is preserved. Full training trajectory plots have been added to Appendix F.2.
>
> ### Case High (where the unique high-variance component is effective)
>
> | (SNR = 1) | p=-2.0 | p=-1.5 | p=-1.0 | p=-0.5 | p=0.0 |
> | :--- | :---: | :---: | :---: | :---: | :---: |
> | **Train** | 0.4924 ± 0.0765 | 0.4959 ± 0.0815 | 0.4855 ± 0.0593 | 0.4902 ± 0.0429 | 0.4993 ± 0.0435 |
> | **Test** | 1.0299 ± 0.0427 | 0.7119 ± 0.0640 | 0.6058 ± 0.0539 | 0.5561 ± 0.0456 | 0.5463 ± 0.0403 |
>
> | (SNR = 2) | p=-2.0 | p=-1.5 | p=-1.0 | p=-0.5 | p=0.0 |
> | :--- | :---: | :---: | :---: | :---: | :---: |
> | **Train** | 0.1882 ± 0.0635 | 0.1902 ± 0.0489 | 0.1977 ± 0.0369 | 0.1957 ± 0.0302 | 0.1974 ± 0.0300 |
> | **Test** | 0.6027 ± 0.1744 | 0.2925 ± 0.0547 | 0.2484 ± 0.0365 | 0.2238 ± 0.0255 | 0.2154 ± 0.0203 |
>
> | (SNR = 3) | p=-2.0 | p=-1.5 | p=-1.0 | p=-0.5 | p=0.0 |
> | :--- | :---: | :---: | :---: | :---: | :---: |
> | **Train** | 0.0942 ± 0.0373 | 0.0973 ± 0.0309 | 0.0986 ± 0.0189 | 0.0996 ± 0.0182 | 0.0993 ± 0.0182 |
> | **Test** | 0.4161 ± 0.1701 | 0.1570 ± 0.0386 | 0.1251 ± 0.0223 | 0.1147 ± 0.0162 | 0.1081 ± 0.0123 |
>
> | (SNR = 4) | p=-2.0 | p=-1.5 | p=-1.0 | p=-0.5 | p=0.0 |
> | :--- | :---: | :---: | :---: | :---: | :---: |
> | **Train** | 0.0553 ± 0.0245 | 0.0566 ± 0.0182 | 0.0584 ± 0.0117 | 0.0588 ± 0.0109 | 0.0580 ± 0.0111 |
> | **Test** | 0.3263 ± 0.1825 | 0.0951 ± 0.0250 | 0.0755 ± 0.0169 | 0.0686 ± 0.0109 | 0.0634 ± 0.0076 |
>
> | (SNR = 5) | p=-2.0 | p=-1.5 | p=-1.0 | p=-0.5 | p=0.0 |
> | :--- | :---: | :---: | :---: | :---: | :---: |
> | **Train** | 0.0369 ± 0.0181 | 0.0382 ± 0.0128 | 0.0385 ± 0.0086 | 0.0383 ± 0.0071 | 0.0383 ± 0.0078 |
> | **Test** | 0.2868 ± 0.1941 | 0.0656 ± 0.0184 | 0.0516 ± 0.0138 | 0.0455 ± 0.0078 | 0.0419 ± 0.0054 |
>
> ### Case Low (where the unique low-variance component is effective)
>
> | (SNR = 1) | p=-2.0 | p=-1.5 | p=-1.0 | p=-0.5 | p=0.0 |
> | :--- | :---: | :---: | :---: | :---: | :---: |
> | **Train** | 0.4850 ± 0.1130 | 0.4648 ± 0.1011 | 0.4716 ± 0.1303 | 0.4953 ± 0.1584 | 0.4820 ± 0.0620 |
> | **Test** | 0.5151 ± 0.0752 | 0.5144 ± 0.0652 | 0.5698 ± 0.1001 | 0.6649 ± 0.1610 | 0.9522 ± 0.1376 |
>
> | (SNR = 2) | p=-2.0 | p=-1.5 | p=-1.0 | p=-0.5 | p=0.0 |
> | :--- | :---: | :---: | :---: | :---: | :---: |
> | **Train** | 0.1965 ± 0.0561 | 0.1840 ± 0.0446 | 0.1844 ± 0.0464 | 0.1940 ± 0.0715 | 0.1962 ± 0.0583 |
> | **Test** | 0.2098 ± 0.0421 | 0.2107 ± 0.0362 | 0.2430 ± 0.0525 | 0.2926 ± 0.1000 | 0.6490 ± 0.2315 |
>
> | (SNR = 3) | p=-2.0 | p=-1.5 | p=-1.0 | p=-0.5 | p=0.0 |
> | :--- | :---: | :---: | :---: | :---: | :---: |
> | **Train** | 0.0969 ± 0.0247 | 0.0974 ± 0.0255 | 0.0979 ± 0.0227 | 0.0959 ± 0.0319 | 0.0972 ± 0.0415 |
> | **Test** | 0.1067 ± 0.0216 | 0.1137 ± 0.0245 | 0.1337 ± 0.0314 | 0.1539 ± 0.0533 | 0.4733 ± 0.2172 |
>
> | (SNR = 4) | p=-2.0 | p=-1.5 | p=-1.0 | p=-0.5 | p=0.0 |
> | :--- | :---: | :---: | :---: | :---: | :---: |
> | **Train** | 0.0585 ± 0.0149 | 0.0588 ± 0.0141 | 0.0585 ± 0.0113 | 0.0583 ± 0.0181 | 0.0585 ± 0.0310 |
> | **Test** | 0.0654 ± 0.0145 | 0.0706 ± 0.0153 | 0.0795 ± 0.0128 | 0.0966 ± 0.0332 | 0.3801 ± 0.1975 |
>
> | (SNR = 5) | p=-2.0 | p=-1.5 | p=-1.0 | p=-0.5 | p=0.0 |
> | :--- | :---: | :---: | :---: | :---: | :---: |
> | **Train** | 0.0383 ± 0.0097 | 0.0384 ± 0.0087 | 0.0383 ± 0.0066 | 0.0378 ± 0.0107 | 0.0374 ± 0.0220 |
> | **Test** | 0.0432 ± 0.0091 | 0.0476 ± 0.0095 | 0.0531 ± 0.0059 | 0.0650 ± 0.0211 | 0.3312 ± 0.1890 |

---

> ### Author Response · Authors · 2025-11-29
> **Response to Reviewer xAve**
>
> ## Regarding the Stochastic Nature of Training (W3 ii)
>
> We appreciate the reviewer’s concern regarding the stochastic nature of training.
>
> In our experiments, we use a full-batch setting, so each training trajectory is deterministic. Extending our theoretical framework to explicitly cover stochastic optimization is an important direction, and we consider it future work.

---

> ### Author Response · Authors · 2025-11-29
> **Response to Reviewer xAve**
>
> ## Regarding The Extensibility of The Single-Index Teacher Setting (W4 i)
>
> We appreciate the reviewer's comment regarding the extensibility of our analysis. We would like to clarify the scope of our theoretical claims and their applicability to multi-index settings.
>
> First, we respectfully clarify that Theorems 1 and 2 describe the information bottleneck mechanism—specifically, how the preconditioner constrains information transfer via the Gram matrix. This derivation depends solely on the interaction between the preconditioner and the input geometry and is entirely independent of the label generation process (i.e. the teacher model). **Therefore, the validity of these theorems holds universally, regardless of whether the task is single-index or multi-index.**
>
> Second, the fundamental mechanism driving generalization remains unchanged when moving from single-index to multi-index settings. The core insight is that learning is facilitated when the spectral components emphasized by the preconditioner ($p$) overlap with the components containing the teacher signal. **In a multi-index setting, the target of this alignment simply shifts from a "single vector" to a "subspace."**
>
> Furthermore, we consider the MNIST experiments in Section 4.2 to be direct empirical validation of this multi-index applicability. **Digit classification inherently relies on a multi-dimensional feature subspace rather than a single direction.** The fact that our framework successfully predicts and controls feature learning in this complex setting provides strong evidence that our insights are not limited to the single-index regime but function robustly in multi-dimensional contexts.
>
> We have incorporated this discussion into Section 3.3 of the revised manuscript to explicitly address the extension to multi-index teachers.
>
> ## Regarding The Generality of The Discussion on First-layer Preconditioning (W4 ii)
>
> We appreciate this thoughtful observation.
>
> While our theoretical derivation focuses on the first layer for tractability, we argue—supported by our empirical results—that the spectral bias established in the first layer remains the dominant factor in deep networks.
>
> **The first layer is the unique interface with the raw input data $X$.** We argue that information filtered out by the preconditioner at this stage is effectively lost and cannot be recovered by subsequent layers, regardless of network depth. Consequently, the spectral bias established in the first layer acts as the dominant factor determining the learnable features for the entire network.
> This concern is directly addressed by our empirical results. In Sections 4.1.2 and 4.2, we employed optimizers such as K-FAC, Sophia-H, and AdaHessian, which apply preconditioning to all layers. Despite this, the observed behaviors strictly followed the predictions of our first-layer theoretical analysis. This provides strong evidence that the spectral bias induced at the input layer does not "wash out" with depth but remains a controlling factor.
>
> Our findings do not contradict Zhang et al. (2025), who highlight the importance of layer-wise preconditioning. Instead, we position our work as complementary: while layer-wise preconditioning is beneficial, we demonstrate that the first layer's preconditioning plays the distinct and decisive role in determining which input information is selected or discarded during feature learning.

---

> ### Author Response · Authors · 2025-11-29
> **Response to Reviewer xAve**
>
> ## Regarding The Experimental Design Ambiguities (W5 i)
>
> **Regarding the OOD experimental procedure (Sec. 4.2.1):**
>
> We confirm that the reviewer’s interpretation (points (a) and (b)) is correct. In our MNIST experiments, we treat digit features as high-variance components and class-wise Gaussian noise as low-variance components. During training, both are perfectly correlated with the class labels, allowing the model to achieve high accuracy by relying on either or both. At test time, we randomize the correspondence between the class labels and one of these features (either the digit or the noise) to render it totally non-predictive. This procedure explicitly allows us to control which component serves as the invariant feature and which as the spurious one. We have revised the explanation in Section 4.2 to clarify this protocol.
>
> **Regarding noise types and strength:**
>
> We specifically selected Gaussian noise for theoretical rigor. Our study focuses on feature selection based on covariance eigenvalues. Gaussian noise allows for precise mathematical control over these eigenvalues, making it the optimal choice for verifying our spectral theory. Other noise types (e.g., Salt-and-Pepper) introduce complex, undefined spectral structures that would obscure the correspondence with our theoretical framework. Furthermore, the noise strength was deliberately set to ensure its variance remains smaller than that of the digit features; significantly increasing this strength would violate the experimental premise of distinguishing between high and low variance components.
>
> **Regarding error bars:**
>
> We apologize if the error bars in Figure 2 were difficult to visualize. They are present in the plots, but due to the high stability of the results across random seeds, the variance is very small, making the bars appear negligible. For precise verification, we have reported the full mean and standard deviation values in Table 2 of Appendix F.2, which we now reference more prominently.

---

> ### Author Response · Authors · 2025-11-29
> **Response to Reviewer xAve**
>
> ## Regarding The Analysis of Related Work (W5 ii)
>
> Thank you for raising concerns about our positioning relative to related work.
>
> We fully acknowledge that Zhang et al. (2025) theoretically demonstrated that layer-wise preconditioning, such as K-FAC, promotes feature learning, and Zhang et al. (2024) suggested that whitening is effective for transfer learning. **In contrast, our primary contribution extends beyond merely demonstrating the effectiveness of specific optimizers; rather, we provide a unified perspective showing that the spectral bias of feature learning—whether it emphasizes high-variance, low-variance, or remains flat—is controllably determined by a single parameter, the preconditioner exponent $p$.** Specifically, while Zhang et al. (2024,2025) identified that whitening ($p=-1$) benefits transfer learning, our framework explains why: from a spectral bias perspective, a flat bias avoids discarding specific frequency components in the source task, thereby preserving the adaptability required for unknown target tasks.
>
> We have revised the conclusion of Section 4.3 to explicitly clarify this deeper relationship and the distinct positioning of our contributions.
>
> ## Regarding The Code Issues (W5 iii)
>
> We sincerely apologize for the usability issues in our initial code submission and deeply appreciate the time you took to test it.
>
> We have fully resolved the reported bugs, including the TypeError in AdaHessian.step() and the data iterator malfunctions. Furthermore, to address the concern regarding insufficient output context, we have upgraded the codebase to automatically save the complete training history and experiment configurations to a user-specified output directory, rather than merely printing single loss values. With these corrections and enhancements, the repository is now fully functional and capable of rigorously reproducing the results reported in the paper.

---

> ### Author Response · Authors · 2025-11-29
> **Response to Reviewer xAve**
>
> ## Regarding the mathematical presentation (W6):
>
> We appreciate the reviewer's feedback regarding the readability of the mathematical sections.
>
> We fully recognize that the theoretical derivations can be dense; therefore, we have explicitly designed the manuscript structure to bridge the gap between mathematical rigor and conceptual clarity. Specifically, we introduced the "Implication" blocks (Implications 1, 2, and 3) immediately following the formal theorems in Section 3. These blocks serve to translate the abstract mathematical results (regarding Gram matrix geometry) into intuitive, plain-text explanations of how they dictate feature learning and generalization. We believe this structure, combined with relegating the detailed proofs to the Appendix, effectively balances the necessity for formal rigor with the need for an accessible, intuitive narrative.

---

> ### Author Response · Authors · 2025-11-29
> **Response to Reviewer xAve**
>
> ## Regarding Why SAM Corresponds to $p = 1$ (Q1)
>
> Thank you for pointing out the derivation of why SAM corresponds to p = 1.
> It can be associated with p=1 through the following simple derivation.
>
> Recall the gradient computation in SAM, denoted as $\boldsymbol{g}_{\text{SAM}}$, which involves computing the gradient at a perturbed parameter location to maximize the loss locally:
>
> \begin{equation}
> \boldsymbol{g}_{\text{SAM}} = \nabla\mathcal{L}\left(\boldsymbol{\theta} + \frac{\rho}{||\nabla\mathcal{L}(\boldsymbol{\theta})||}\nabla\mathcal{L}(\boldsymbol{\theta})\right),
> \end{equation}
>
> where $\rho$ is the neighborhood size. Let us define the perturbation vector $\boldsymbol{\delta}$ as:
>
> \begin{equation}
> \boldsymbol{\delta} = \frac{\rho}{||\nabla\mathcal{L}(\boldsymbol{\theta})||}\nabla\mathcal{L}(\boldsymbol{\theta}).
> \end{equation}
>
> Assuming that $\rho$ is sufficiently small, we can apply a first-order Taylor expansion of the gradient $\nabla\mathcal{L}(\boldsymbol{\theta} + \boldsymbol{\delta})$ around $\boldsymbol{\theta}$. Let $\boldsymbol{H} = \nabla^2\mathcal{L}(\boldsymbol{\theta})$ denote the Hessian matrix of the loss function at $\boldsymbol{\theta}$. The approximation proceeds as follows:
>
> \begin{align}
> \boldsymbol{g}_{\text{SAM}} &\approx \nabla\mathcal{L}(\boldsymbol{\theta}) + \boldsymbol{H} \boldsymbol{\delta} \nonumber = \nabla\mathcal{L}(\boldsymbol{\theta}) + \boldsymbol{H} \left( \frac{\rho}{||\nabla\mathcal{L}(\boldsymbol{\theta})||}\nabla\mathcal{L}(\boldsymbol{\theta}) \right) \nonumber = \nabla\mathcal{L}(\boldsymbol{\theta}) + \frac{\rho}{||\nabla\mathcal{L}(\boldsymbol{\theta})||} \boldsymbol{H} \nabla\mathcal{L}(\boldsymbol{\theta}).
> \end{align}
>
> By factoring out the gradient term $\nabla\mathcal{L}(\boldsymbol{\theta})$, we obtain the final approximate form:
>
> \begin{equation}
> \boldsymbol{g}_{\text{SAM}} \approx \left( I + \frac{\rho}{||\nabla\mathcal{L}(\boldsymbol{\theta})||} \boldsymbol{H} \right) \nabla\mathcal{L}(\boldsymbol{\theta}).
> \end{equation}
>
> This derivation illustrates that the effective gradient used in SAM is approximately equivalent to the standard gradient preconditioned by the matrix $\left( I + \frac{\rho}{||\nabla\mathcal{L}(\boldsymbol{\theta})||} \boldsymbol{H} \right)$. Here, since the first-layer neuron-wise Hessian $\boldsymbol{H}$ can be approximated by the input covariance $\boldsymbol{\Sigma}_X$ (see Appendix C), SAM corresponds to $p = 1$ in our formulation.
>
> We have also included this derivation in the appendix.
>
> ## Regarding How to Aggregate Results across Multiple Seeds (Q2)
>
> Thank you for your thoughtful feedback and constructive suggestions.
>
> We clarify that for each experimental configuration, we performed 10 independent training runs using distinct random seeds to initialize the models. In the figures presented in the paper, the solid lines represent the mean across these 10 runs, while the shaded areas (or error bars) indicate the standard deviation, reflecting the variability due to initialization.
>
> ## Regrading Transfer Learning Experiments on Real-world Data (Q3)
>
> Thank you for highlighting the importance of real‑world transfer learning experiments.
>
> We acknowledge the importance of large-scale benchmarks like ImageNet. However, we face the same fundamental challenge described in our response regarding Section 4.1: in complex real-world datasets, the spectral location of the true teacher signal is unknown. This makes it impossible to rigorously verify whether any performance gain stems from the spectral alignment mechanism we propose. Consequently, we prioritized establishing the theoretical foundation in controllable settings first. We view the development of an analytical framework to identify and exploit these signal components in real-world pretraining as a vital direction for our future work.

---

> ### Author Response · Authors · 2025-11-29
> **Response to Reviewer xAve**
>
> ## Regarding Practical Methods for Aligning Spectral Bias with The Teacher (Q4)
>
> We appreciate the question on practical methods to align an optimizer’s spectral bias with the teacher signal.
>
> To demonstrate the practical implications of our framework, we explore a method for selecting the optimal optimizer$-$specifically in terms of generalization performance$-$prior to training. Drawing inspiration from [1], we propose estimating which spectral components are effective for label prediction using the inference of the model at initialization.
>
> [1] defined the Initial Alignment (INAL) as the expected squared correlation between the target function $f^*$ and the output of the neural network at initialization $f_{\theta_0}$ (More precisely, they calculate the following INAL for each neuron and take the maximum value.:
>
> \begin{equation}
> \text{INAL}(f^\star, f_{\theta_0}) = \mathbb{E}[\langle f^\star, f_{\theta_0} \rangle^2].
> \end{equation}
>
> They established that a higher INAL serves as a strong indicator of learnability and subsequent generalization performance, signifying that the model at initialization already possesses components effective for label prediction.
>
> Extending this concept to our spectral analysis, we calculate the INAL \textit{separately} for each eigen-component of the input to identify specific important directions. We applied this method to the synthetic datasets (\setting{High} and \setting{Low} cases) described in the main text. The results are summarized in Tables below.
>
> In both cases, we observed a distinct spike in the INAL value relative to the median specifically for the components carrying the informative signal (Component 0 for Case High, and Component 9 for Case Low).
>
> *1. High (where the unique high-variance component is effective)*
>
> | Component | INAL  | Relative ratio (vs. median) |
> | :--- | :---: | :---: |
> | **0** | **0.8117 ± 1.1558** | **1.95×** |
> | 1 | 0.4377 ± 0.6016 | 1.05× |
> | 2 | 0.4208 ± 0.5996 | 1.01× |
> | 3 | 0.4383 ± 0.6039 | 1.05× |
> | 4 | 0.4156 ± 0.5919 | 1.00× |
> | 5 | 0.4259 ± 0.6095 | 1.02× |
> | 6 | 0.3967 ± 0.5974 | 0.95× |
> | 7 | 0.4140 ± 0.5842 | 1.00× |
> | 8 | 0.4060 ± 0.5724 | 0.98× |
> | 9 | 0.4149 ± 0.5833 | 1.00× |
>
> *2. Low (where the unique low-variance component is effective)*
>
> | Component | INAL (mean ± std) | Relative ratio (vs. median) |
> | --- | :---: | :---: |
> | 0 | 0.01783 ± 0.02449 | 1.06× |
> | 1 | 0.01686 ± 0.02330 | 1.01× |
> | 2 | 0.01652 ± 0.02358 | 0.99× |
> | 3 | 0.01608 ± 0.02187 | 0.96× |
> | 4 | 0.01668 ± 0.02442 | 1.00× |
> | 5 | 0.01641 ± 0.02342 | 0.98× |
> | 6 | 0.01810 ± 0.02464 | 1.08× |
> | 7 | 0.01728 ± 0.02398 | 1.03× |
> | 8 | 0.01621 ± 0.02407 | 0.97× |
> | **9** | **0.03035 ± 0.04174** | **1.81×** |
>
>
> These results suggest that task-relevant spectral components can be estimated in a training-free manner. We believe this validates our framework's potential to guide the selection of the optimal optimizer (i.e., the optimal preconditioner power $p$) by matching the spectral bias to the task structure, thereby replacing heuristic choices with a principled, task-aligned approach.
>
> We added this results and discussion to the appendix in our manuscript.
>
> [1] Abbe, Emmanuel, et al. "An initial alignment between neural network and target is needed for gradient descent to learn." International Conference on Machine Learning. PMLR, 2022.

---

> ### Author Response · Authors · 2025-11-29
> **Note to the AC**
>
> **There is no response to Weakness 2 because the reviewer did not list one (they went from Weakness 1 directly to Weakness 3), and we fully addressed all concerns.**

---

### Meta-Review · Area_Chair_twPu · 2026-01-05

**Summary:**

This work shows how training with a preconditioning matrix leads to different types of feature learning. The paper is solid, but I find its novelty and significance rather lacking. That preconditioning matrix changes how a feature is learned is known in previous works. On the technical side, this theory is rather straightforward and not so insightful. I think this paper is incremental (for example, one reviewer said the contribution is "not surprising at all and relatively shallow") and truly borderline. I do not mind accepting this paper, but I lean towards rejection.

In some sense, another problem with the paper is that it tries to combine theory with experiments, but neither part stands out as sufficiently important. As a theory paper, the theory is not so novel; as an experimental paper, the evaluation is too minimal (as pointed out by some reviewers).

**Reviewer Concerns:**

Some additional experiments are presented, but they do not go beyond the limited scope of the manuscript (mostly synthetic or small problems)

The main problem, which is the limited novelty, does not feel sufficiently well addressed.

**Reviewer Scores:**

I am not sure how to answer this question... I think xAve has some reasonable probability of raising score from 4 to 6, (let's say 50% probability), given the effort the authors put into the revision. But this still makes the paper borderline

---

### Decision · Program_Chairs · 2026-01-26

Reject